# On the Power of Pre-training for Generalization in RL: Provable Benefits and Hardness

## Abstract

Generalization in Reinforcement Learning (RL) aims to train an agent during training that generalizes to the target environment. In this work, we first point out that RL generalization is fundamentally different from the generalization in supervised learning, and fine-tuning on the target environment is necessary for good test performance. Therefore, we seek to answer the following question: how much can we expect pre-training over training environments to be helpful for efficient and effective fine-tuning? On one hand, we give a surprising result showing that asymptotically, the improvement from pre-training is at most a constant factor. On the other hand, we show that pre-training can be indeed helpful in the non-asymptotic regime by designing a policy collection-elimination (PCE) algorithm and proving a distribution-dependent regret bound that is independent of the state-action space. We hope our theoretical results can provide insight towards understanding pre-training and generalization in RL.

## 1    Introduction

Reinforcement learning (RL) is concerned with sequential decision making problems in which the agent interacts with the environment aiming to maximize its cumulative reward. This framework has achieved tremendous successes in various fields such as game playing (Mnih et al., 2013; Silver et al., 2017; Vinyals et al., 2019), resource management (Mao et al., 2016), recommendation systems (Shani et al., 2005; Zheng et al., 2018) and online advertising (Cai et al., 2017). However, many empirical applications of RL algorithms are typically restricted to the single environment setting. That is, the RL policy is learned and evaluated in the exactly same environment. This learning paradigm can lead to the issue of overfitting in RL (Sutton, 1995; Farebrother et al., 2018), and may have degenerate performance when the agent is deployed to an unseen (but similar) environment.

The ability to generalize to test environments is important to the success of reinforcement learning algorithms, especially in the real applications such as autonomous driving (Shalev-Shwartz et al., 2016; Sallab et al., 2017), robotics (Kober et al., 2013; Kormushev et al., 2013) and health care (Yu et al., 2021). In these real-world tasks, the environment can be dynamic, open-ended and always changing. We hope the agent can learn meaningful skills in the training stage and be robust to the variation during the test stage. Furthermore, in applications such as robotics where we have a simulator to efficiently and safely generate unlimited data, we can firstly train the agent in the randomized simulator models and then generalize it to the real environment (Rusu et al., 2017; Peng et al., 2018; Andrychowicz et al., 2020). An RL algorithm with good generalization ability can greatly reduce the demand of real-world data and improve test-time performance.

Generalization in supervised learning has been widely studied for decades (Mitchell et al., 1986; Bousquet & Elisseeff, 2002; Kawaguchi et al., 2017). For a typical supervised learning task such as classification, given a hypothesis space $\mathcal{H}$ and a loss function $\ell$, the agent aims to find an optimal solution in the average manner. That is, we hope the solution is near-optimal compared with the optimal hypothesis $h^*$ *in expectation* over the data distribution, which is formally defined as $h^* = \arg\min_{h \in \mathcal{H}} \mathbb{E}\big[\ell(h(X), Y)\big]$. From this perspective, generalization in RL is fundamentally different. Once the agent is deployed in the test environment $\mathcal{M}$ sampled from distribution $\mathbb{D}$, it is expected to achieve comparable performance with the optimal policy in $\mathcal{M}$. In other words, we hope the learned policy can perform near-optimal compared with the optimal value $V_{\mathcal{M}}^*$ *in instance* for the sampled test environment $\mathcal{M}$.

Unfortunately, as discussed in many previous works (Malik et al., 2021; Ghosh et al., 2021), the instance-optimal solution in the target environment can be statistically intractable without additional assumptions. We formulate this intractability into a lower bound (Proposition 1) to show that it is impractical to directly obtain a near-optimal policy for the test environment $\mathcal{M}^*$ with high probability. This motivates us to ask: *in what settings can the generalization problem in RL be tractable?*

Targeting on RL generalization, the agent is often allowed to further interact with the test environment to improve its policy. For example, many previous results in robotics have demonstrated that fine-tuning in the test environment can greatly improve the test performance for sim-to-real transfer (Rusu et al., 2017; James et al., 2019; Rajeswaran et al., 2016). Therefore, one possible way to formulate generalization is to allow further interaction with the target environment during the test stage. Specifically, suppose the agent interacts with MDP $\mathcal{M} \sim \mathbb{D}$ in the test stage, and we measure the performance of the fine-tuning algorithm $\mathcal{A}$ using the expected regret in $K$ episodes, i.e. $\mathrm{Reg}_K(\mathbb{D}, \mathcal{A}) = \mathbb{E}_{\mathcal{M} \sim \mathbb{D}} \left[ \sum_{k=1}^K \left( V_{\mathcal{M}}^{\pi^*(\mathcal{M})} - V_{\mathcal{M}}^{\pi_k} \right) \right]$. In this setting, can the information obtained from pre-training [1] help reduce the regret suffered during the test stage?

In addition, when the test-time fine-tuning is not allowed, what can we expect the pre-training to be helpful? As discussed above, we can no longer demand instance-optimality in this setting, but can only step back and pursue a near-optimal policy *in expectation*. Specifically, our goal is to perform near-optimal in terms of the optimal policy with maximum value in expectation, i.e. $\pi^*(\mathbb{D}) = \arg\max_{\pi \in \Pi} \mathbb{E}_{\mathcal{M} \sim \mathbb{D}} V_{\mathcal{M}}^{\pi}$. Here $V_{\mathcal{M}}^{\pi}$ is the value function of the policy $\pi$ in MDP $\mathcal{M}$. We seek to answer: is it possible to design a sample-efficient training algorithm that returns a $\epsilon$-optimal policy $\pi$ *in expectation*, i.e. $\mathbb{E}_{\mathcal{M} \sim \mathbb{D}} \left[ V_{\mathcal{M}}^{\pi^*(\mathbb{D})} - V_{\mathcal{M}}^{\pi} \right] \leq \epsilon$?

**Main contributions.** In this paper, we theoretically study RL generalization in the above two settings. Our contributions can be summarized as follows:

• When fine-tuning is allowed, we study the benefit of pre-training for the test-time performance. Since all information we can gain from training is no more than the distribution $\mathbb{D}$ itself, we start with a somewhat surprising theorem showing the limitation of this benefit: there exists hard cases where, even if the agent has *exactly* learned the environment distribution $\mathbb{D}$ in the training stage, it *cannot improve* the test-time regret up to a universal factor in the asymptotic setting ($K \to \infty$). In other words, knowing the distribution $\mathbb{D}$ cannot provide more information in consideration of the regret asymptotically. Our theorem is proved by using Radon transform and Lebesgue integral analysis to give a global level information limit, which we believe are novel techniques for RL communities.

• Inspired by this lower bound, we focus on the non-asymptotic setting, and study whether and how much we can reduce the regret in this case. We propose an efficient pre-training and test-time fine-tuning algorithm called PCE (Policy Collection-Elimination). By maintaining a minimum policy set that generalizes well, it achieves a regret upper bound $\tilde{\mathcal{O}}\left(\sqrt{\mathcal{C}(\mathbb{D})K}\right)$ in the test stage, where $\mathcal{C}(\mathbb{D})$ is a complexity measure of the distribution $\mathbb{D}$. This bound removes the polynomial dependence on the cardinality of state-action space by leveraging the information obtained from pre-training. We give a fine-grained analysis on the value of $\mathcal{C}(\mathbb{D})$ and show that our bound can be significantly smaller than state-action space dependent bound in many settings.

• When the agent cannot interact with the test environment, we propose an efficient algorithm called OMERM (Optimistic Model-based Empirical Risk Minimization) to find a near-optimal policy *in expectation*. This algorithm is guaranteed to return a $\epsilon$-optimal policy with $\mathcal{O}\left(\log\left(\mathcal{N}_{\epsilon/(12H)}^{\Pi}\right)/\epsilon^2\right)$ sampled MDP tasks in the training stage where $\mathcal{N}_{\epsilon/(12H)}^{\Pi}$ is the complexity of the policy class. This rate matches the traditional generalization rate in many supervised learning results (Mohri et al., 2018; Kawaguchi et al., 2017).

## 2 RELATED WORKS

**Generalization and Multi-task RL.** Many empirical works study how to improve generalization for deep RL algorithms (Packer et al., 2018; Zhang et al., 2020; Ghosh et al., 2021). We refer readers

---

[1] We call the training stage "pre-training" when interactions with the test environment are allowed.

to a recent survey Kirk et al. (2021) for more discussion on empirical results. Our paper is more closely related to the recent works towards understanding RL generalization from the theoretical perspective.Wang et al. (2019) focused on a special class of reparameterizable RL problems, and derive generalization bounds based on Rademacher complexity and the PAC-Bayes bound. Malik et al. (2021); Duan et al. (2021) also provided lower bounds showing that instance-optimal solution is statistically difficult for RL generalization when we cannot access the sampled test environment. Further, they proposed efficient algorithms which is guaranteed to return a near-optimal policy for deterministic MDPs under the strong proximity condition they introduced. Our paper is also related to recent works studying multi-task learning in RL (Tirinzoni et al., 2020; Hu et al., 2021; Zhang & Wang, 2021; Lu et al., 2021), in which they studied how to transfer the knowledge learned from previous tasks to new tasks. Their problem formulation is different from ours since they study the multi-task setting where the MDP is selected from a given MDP set without probability mechanism(Brunskill & Li, 2013). In addition, they typically assume that all the tasks have similar transition dynamics or share common representations.

**Provably Efficient Exploration in RL.** Recent years have witnessed many theoretical results studying provably efficient exploration in RL (Osband et al., 2013; Azar et al., 2017; Osband & Van Roy, 2017; Jin et al., 2018; 2020b; Wang et al., 2020; Zhang et al., 2021) with the minimax regret for tabular MDPs with non-stationary transition being $\tilde{\mathcal{O}}(\sqrt{HSAK})$. These results indicate that polynomial dependence on the whole state-action space is unavoidable without additional assumptions. Their formulation corresponds to the single-task setting where the agent only interacts with a single environment aiming to maximize its cumulative rewards without pre-training. The regret defined in the fine-tuning setting coincides with the concept of Bayesian regret in the previous literature (Osband et al., 2013; Osband & Van Roy, 2017; O'Donoghue, 2021). The best-known Bayesian regret for tabular RL is $\tilde{\mathcal{O}}(\sqrt{HSAK})$ when applied to our setting (O'Donoghue, 2021).

## 3 PRELIMINARY AND FRAMEWORK

**Notations**  Throughout the paper, we use $[N]$ to denote the set $\{1, \cdots, N\}$ where $N \in \mathbb{N}_+$. For an event $\mathcal{E}$, let $\mathbb{I}[\mathcal{E}]$ be the indicator function of event $\mathcal{E}$, i.e. $\mathbb{I}[\mathcal{E}] = 1$ if and only if $\mathcal{E}$ is true. For any domain $\Omega$, we use $C(\Omega)$ to denote the continuous function on $\Omega$. We use $\mathcal{O}(\cdot)$ to denote the standard big $O$ notation, and $\tilde{\mathcal{O}}(\cdot)$ to denote the big $O$ notation with $\log(\cdot)$ term omitted.

### 3.1 EPISODIC MDPS

An episodic MDP $\mathcal{M}$ is specified as a tuple $(\mathcal{S}, \mathcal{A}, \mathbb{P}_{\mathcal{M}}, R_{\mathcal{M}}, H)$, where $\mathcal{S}, \mathcal{A}$ are the state and action space with cardinality $S$ and $A$ respectively, and $H$ is the steps in one episode. $\mathbb{P}_{\mathcal{M},h} : \mathcal{S} \times A \mapsto \Delta(\mathcal{S})$ is the transition such that $\mathbb{P}_{\mathcal{M},h}(s'|s, a)$ denotes the probability to transit to state $s'$ if action $a$ is taken in state $s$ in step $h$. $R_{\mathcal{M},h} : \mathcal{S} \times \mathcal{A} \mapsto \Delta(\mathbb{R})$ is the reward function such that $R_{\mathcal{M},h}(s, a)$ is the distribution of reward with non-negative mean $r_{\mathcal{M},h}(s, a)$ when action $a$ is taken in state $s$ at step $h$. In order to compare with traditional generalization, we make the following assumption:

**Assumption 1.** *The total mean reward is bounded by 1, i.e. $\forall \mathcal{M} \in \Omega, \sum_{h=1}^{H} r_{\mathcal{M},h}(s_h, a_h) \leq 1$ for all trajectory $(s_1, a_1, \cdots, s_H, a_H)$ with positive probability in $\mathcal{M}$; The reward mechanism $R_{\mathcal{M}}(s, a)$ is 1-subgaussian, i.e. $\mathbb{E}_{X \sim R_{\mathcal{M},h}(s,a)}[\exp(\lambda[X - r_{\mathcal{M},h}(s, a)])] \leq \exp \frac{\lambda^2}{2}$ for all $\lambda \in \mathbb{R}$.*

The total reward assumption follows the previous works on horizon-free RL (Ren et al., 2021; Zhang et al., 2021; Li et al., 2022) and covers the traditional setting where $r_{\mathcal{M},h}(s, a) \in [0, 1]$ by scaling $H$, and it is more natural in environments with sparse rewards (Vecerik et al., 2017; Riedmiller et al., 2018). In addition, it allows us to compare with supervised learning bound where $H = 1$ and the loss is bounded by $[0, 1]$. The subgaussian assumption is more common in practice and is widely used in bandits (Lattimore & Szepesvári, 2020). It also covers traditional RL setting where $R_{\mathcal{M},h}(s, a) \in \Delta([0, 1])$, and allows us to study MDP environment with a wider range. For the convenience of explanation, we assume the agent always starts from the same state $s_1$. It is straightforward to recover the initial state distribution $\mu$ from this setting by adding an initial state $s_0$ with transition $\mu$ (Du et al., 2019; Chen et al., 2021).

**Policy and Value Function.**  A policy $\pi$ is set of $H$ functions where each maps a state to an action distribution, i.e. $\pi = \{\pi_h\}_{h=1}^{H}, \pi_h : \mathcal{S} \mapsto \Delta(\mathcal{A})$ and $\pi$ can be stochastic. We de-

note the set of all policies described above as $\Pi$. We define $\mathcal{N}_\epsilon^\Pi$ as the $\epsilon$-covering number of the policy space $\Pi$ w.r.t. distance $\mathrm{d}(\pi^1, \pi^2) = \max_{s \in \mathcal{S}, h \in [H]} \|\pi_h^1(\cdot|s) - \pi_h^2(\cdot|s)\|_1$. Given $\pi$ and $h \in [H]$, we define the Q-function $Q_{\mathcal{M},h}^\pi : \mathcal{S} \times \mathcal{A} \mapsto \mathbb{R}_+$, where $Q_{\mathcal{M},h}^\pi(s, a) = r_{\mathcal{M},h}(s, a) + \sum_{s' \in \mathcal{S}} \mathbb{P}_{\mathcal{M},h}(s'|s, a) V_{\mathcal{M},h+1}^\pi(s')$, and the V-function $V_{\mathcal{M},h}^\pi : \mathcal{S} \mapsto \mathbb{R}_+$, where $V_{\mathcal{M},h}^\pi(s) = \mathbb{E}_{a \sim \pi_h(\cdot|s)} Q_{\mathcal{M},h}^\pi(s, a)$ for $h \leq H$ and $V_{\mathcal{M},H+1}^\pi(s) = 0$. We abbreviate $V_{\mathcal{M},1}^\pi(s_1)$ as $V_{\mathcal{M}}^\pi$, which can be interpreted as the value when executing policy $\pi$ in $\mathcal{M}$. Following the notations in previous works, we use $\mathbb{P}_h V(s, a)$ as the shorthand of $\sum_{s' \in \mathcal{S}} \mathbb{P}_h(s'|s, a) V(s')$ in our analysis.

## 3.2 RL Generalization Formulation

We mainly study the setting where all MDP instances we face in training and testing stages are *i.i.d.* sampled from a distribution $\mathbb{D}$ supported on a (possibly infinite) countable set $\Omega$. For an MDP $\mathcal{M} \in \Omega$, we use $\mathcal{P}(\mathcal{M})$ to denote the probability of sampling $\mathcal{M}$ according to distribution $\mathbb{D}$. For an MDP set $\tilde{\Omega} \subseteq \Omega$, we similarly define $\mathcal{P}(\tilde{\Omega}) = \sum_{\mathcal{M} \in \tilde{\Omega}} \mathcal{P}(\mathcal{M})$. We assume that $\mathcal{S}, \mathcal{A}, H$ is shared by all MDPs, while the transition and reward are different. When interacting with a sampled instance $\mathcal{M}$, one does *not* know which instance it is, but can only identify its model through interactions.

In the training (pre-training) stage, the agent can sample *i.i.d.* MDP instances from the unknown distribution $\mathbb{D}$. The overall goal is to perform well in the test stage with the information learned in the training stage. Define the optimal policy as

$$\pi^*(\mathcal{M}) = \arg\max_{\pi \in \Pi} V_{\mathcal{M}}^\pi, \ \pi^*(\mathbb{D}) = \arg\max_{\pi \in \Pi} \mathbb{E}_{\mathcal{M} \sim \mathbb{D}} V_{\mathcal{M}}^\pi.$$

We say a policy $\pi$ is $\epsilon$-optimal *in expectation*, if $\mathbb{E}_{\mathcal{M} \sim \mathbb{D}}\left[V_{\mathcal{M}}^{\pi^*(\mathbb{D})} - V_{\mathcal{M}}^\pi\right] \leq \epsilon$. We say a policy $\pi$ is $\epsilon$-optimal *in instance*, if $\mathbb{E}_{\mathcal{M} \sim \mathbb{D}}\left[V_{\mathcal{M}}^{\pi^*(\mathcal{M})} - V_{\mathcal{M}}^\pi\right] \leq \epsilon$.

**Without Test-time Interaction.** When the interaction with the test environment is unavailable, optimality *in instance* can be statistically intractable, and we can only pursue optimality *in expectation*. We formulate this difficulty into the following proposition.

**Proposition 1.** *There exists an MDP support $\Omega$, such that for any distribution $\mathbb{D}$ with positive p.d.f. $p(r)$, $\exists \epsilon_0 > 0$, and for any deployed policy $\hat{\pi}$,*

$$\mathbb{E}_{\mathcal{M}^* \sim \mathbb{D}}\left[V_{\mathcal{M}^*}^{\pi^*(\mathcal{M}^*)} - V_{\mathcal{M}^*}^{\hat{\pi}}\right] \geq \epsilon_0.$$

Proposition 1 is proved by constructing $\Omega$ as a set of MDPs with opposed optimal action, and the complete proof can be found in Appendix A. When $\Omega$ is discrete, there exists hard instances where the proposition holds for $\epsilon_0 \geq \frac{1}{2}$. This implies that without test-time interactions or special knowledge on the structure of $\Omega$ and $\mathbb{D}$, it is impractical to be near optimal *in instance*. This intractability arises from the demand on instance optimal policy, which is never asked in supervised learning.

**With Test-time Interaction.** To pursue the optimality *in instance*, we study the problem of RL generalization with test-time interaction. When our algorithm is allowed to interact with the target MDP $\mathcal{M}^* \sim \mathbb{D}$ for $K$ episodes in the test stage, we want to reduce the regret, which is defined as

$$\mathrm{Reg}_K(\mathbb{D}, \mathcal{A}) \triangleq \mathbb{E}_{\mathcal{M}^* \sim \mathbb{D}} \mathrm{Reg}_K(\mathcal{M}^*, \mathcal{A}), \ \mathrm{Reg}_K(\mathcal{M}^*, \mathcal{A}) \triangleq \sum_{k=1}^{K} [V_{\mathcal{M}^*}^{\pi^*(\mathcal{M}^*)} - V_{\mathcal{M}^*}^{\pi_k}],$$

where $\pi_k$ is the policy that $\mathcal{A}$ deploys in episode $k$. Here $\mathcal{M}^*$ is unknown and unchanged during all $K$ episodes. The choice of Bayesian regret is more natural in generalization, and can better evaluate the performance of an algorithm in practice. From the standard Regret-to-PAC technique (Jin et al., 2018; Dann et al., 2017), an algorithm with $\tilde{\mathcal{O}}(\sqrt{K})$ regret can be transformed to an algorithm that returns an $\epsilon$-optimal policy with $\tilde{\mathcal{O}}(1/\epsilon^2)$ trajectories. Therefore, we believe regret can also be a good criterion to measure the sample efficiency of fine-tuning algorithms in the test stage.

## 4 Results for the Setting with Test-time Interaction

In this section, we study the setting where the agent is allowed to interact with the sampled test MDP $\mathcal{M}^*$. When there is no pre-training stage, the typical regret bound in the test stage is

$\tilde{\mathcal{O}}(\sqrt{SAHK})$(Zhang et al., 2021). For generalization in RL, we mainly care about the performance in the test stage, and hope the agent can reduce test regret by leveraging the information learned in the pre-training stage. Obviously, when $\Omega$ is the set of all tabular MDPs and the distribution $\mathbb{D}$ is uniform over $\Omega$, pre-training can do nothing on improving the test regret, since it provides no extra information for the test stage. Therefore, we seek a *distribution-dependent* improvement that is better than traditional upper bound in most of benign settings.

## 4.1 LOWER BOUND

We start by understanding how much information the pre-training stage can provide *at most*. One natural focus is on the MDP distribution $\mathbb{D}$, which is a sufficient statistic of the possible environment that the agent will encounter in the test stage. We strengthen the algorithm by directly telling it the accurate distribution $\mathbb{D}$, and analyze how much this extra information can help to improve the regret. Specifically, we ask: Is there a multiplied factor $\mathcal{C}(\mathbb{D})$ that is small when $\mathbb{D}$ enjoys some benign properties (e.g. $\mathbb{D}$ is sharp and concentrated), such that when knowing $\mathbb{D}$, there exists an algorithm that can reduce the regret by a factor of $\mathcal{C}(\mathbb{D})$ for large enough $K$?

Perhaps surprisingly, our answer towards this question is *negative* for all large enough $K$ in the asymptotic case. As is formulated in Theorem 1, the importance of $\mathbb{D}$ is constrained by a universal factor $c_0$ asymptotically. Here $c_0 = \frac{1}{16}$ holds universally and does not depend on $\mathbb{D}$. This theorem implies that no matter what distribution $\mathbb{D}$ is, for sufficiently large $K$, any algorithm can only reduce the total regret by at most a constant factor with the extra knowledge of $\mathbb{D}$.

**Theorem 1.** *There exists an MDP instance set $\Omega$, a universal constant $c_0 = \frac{1}{16}$, and an algorithm $\hat{\mathcal{A}}$ that only inputs the episode $K$, such that for* any *distribution $\mathbb{D}$ with positive p.d.f. $p(r) \in C(\Omega)$ (which $\hat{\mathcal{A}}$ does NOT know), any algorithm $\mathcal{A}$ that inputs $\mathbb{D}$ and the episode $K$,*

1. *$\Omega$ is not degraded, i.e. $\lim_{K\to\infty} \text{Reg}_K(\mathbb{D}, \mathcal{A}(\mathbb{D}, K)) = +\infty$.*

2. *Knowing the distribution is useless up to a constant, i.e.*

$$\liminf_{K\to\infty} \frac{\text{Reg}_K(\mathbb{D}, \mathcal{A}(\mathbb{D}, K))}{\text{Reg}_K(\mathbb{D}, \hat{\mathcal{A}}(K))} \geq c_0.$$

In Theorem 1, Point (1) avoids any trivial support $\Omega$ where $\exists \pi^*$ that is optimal for all $\mathcal{M} \in \Omega$, in which case the distribution is of course useless since $\hat{\mathcal{A}}$ can be optimal by simply following $\pi^*$ even it does not know $\mathbb{D}$. Note that our bounds hold for any distribution $\mathbb{D}$, which indicates that even a very sharp distribution cannot provide useful information in the asymptotic case where $K \to \infty$. We point out that the value of $c_0$ depends on the coefficient of previous upper and lower bound, and we conjecture that it could be arbitrarily close to 1.

We defer the complete proof to Appendix B, and briefly sketch the intuition here. The key observation is that the information provided in the training stage (prior) is fixed, while the required information gradually increase as $K$ increases. When $K = 1$, the agent can clearly benefit from the knowledge of $\mathbb{D}$. Without this knowledge, all it can do is a random guess since it has never interacted with $\mathcal{M}^*$ before. However, when $K$ is large, the algorithm can interact with $\mathcal{M}^*$ many times and learn $\mathcal{M}^*$ more accurately, while the prior $\mathbb{D}$ will become relatively less informative. As a result, the benefits of knowing $\mathbb{D}$ vanishes eventually.

Theorem 1 lower bounds the improvement of regret by a constant. As is commonly known, the regret bound can be converted into a PAC-RL bound (Jin et al., 2018; Dann et al., 2017). This implies that when $\delta, \epsilon \to 0$, in terms of pursuing a $\epsilon$-optimal policy to $\pi^*(\mathcal{M}^*)$, pre-training cannot help reduce the sample complexity. Despite negative, we point out that this theorem only describe the asymptotic setting where $K \to \infty$, but it imposes no constraint when $K$ is fixed.

## 4.2 NON-ASYMPTOTIC UPPER BOUND

In the last subsection, we provide a lower bound showing that the information obtained from the training stage can be useless in the asymptotic setting where $K \to \infty$. In practice, a near-optimal regret in the non-asymptotic setting is also desirable in many applications. In this section, we fix the value of $K$ and seek to design an algorithm such that it can leverage the pre-training information

and reduce $K$-episode test regret. To avoid redundant explanation for single MDP learning, we formulate the following oracles.

**Definition 1.** *(Policy learning oracle) We define $\mathbb{O}_l(\mathcal{M}, \epsilon, \log(1/\delta))$ as the policy learning oracle which can return a policy $\pi$ that is $\epsilon$-optimal w.r.t. MDP $\mathcal{M}$ with probability at least $1 - \delta$, i.e. $V_{\mathcal{M}}^*(s_1) - V_{\mathcal{M}}^\pi(s_1) \leq \epsilon$. The randomness of the policy $\pi$ is due to the randomness of both the oracle algorithm and the environment.*

**Definition 2.** *(Policy evaluation oracle) We define $\mathbb{O}_e(\mathcal{M}, \pi, \epsilon, \log(1/\delta))$ as the policy evaluation oracle which can return a value $v$ that is $\epsilon$-close to the value function $V_{\mathcal{M}}^\pi(s_1)$ with probability at least $1 - \delta$, i.e. $|v - V_{\mathcal{M}}^\pi(s_1)| \leq \epsilon$. The randomness of the value $v$ is due to the randomness of both the oracle algorithm and the environment.*

Both oracles can be efficiently implemented using the previous algorithms for single-task MDPs. Specifically, we can implement the policy learning oracle using algorithms such as UCBVI (Azar et al., 2017), LSVI-UCB (Jin et al., 2020b) and GOLF (Jin et al., 2021) with polynomial sample complexities, and the policy evaluation oracle can be achieved by the standard Monte Carlo method (Sutton & Barto, 2018).

### 4.2.1 ALGORITHM

There are two major difficulties in designing the algorithm. First, what do we want to learn during the pre-training process and how to learn it? One idea is to directly learn the whole distribution $\mathbb{D}$, which is all that we can obtain for the test stage. However, this requires $\tilde{\mathcal{O}}(|\Omega|^2/\delta^2)$ samples for a required accuracy $\delta$, and is unacceptable when $|\Omega|$ is large or even infinite. Second, how to design the test-stage algorithm to leverage the learned information effectively? If we cannot effectively use the information from the pre-training, the regret or samples required in the test stage can be $\tilde{\mathcal{O}}(\text{poly}(S, A))$ in the worst case.

---

**Algorithm 1** PCE (**P**olicy **C**ollection-**E**limination)

**Pre-training Stage**

1: **Input**: episode number $K$, policy learning oracle $\mathbb{O}_l$ and policy evaluation oracle $\mathbb{O}_e$
2: Initialize: $\delta = \epsilon = 1/\sqrt{K}$, the number of the sampled MDPs $N = \log(1/\delta)/\delta^2$
3: **for** phase $l = 1, \cdots$ **do**
4:      Sample an MDP set $\hat{\Omega}$ with $N$ MDPs $\{\mathcal{M}_1, \mathcal{M}_2, \cdots, \mathcal{M}_N\}$ from distribution $\mathbb{D}$
5:      **for** $j = 1, \cdots, N$ **do**
6:          Calculate $\pi_j = \mathbb{O}_l(\mathcal{M}_j, \epsilon/2, \log(N/\delta))$ for the MDP $\mathcal{M}_j$
7:      **for** $i, j = 1, \cdots, N$ **do**
8:          Calculate $v_{i,j} = \mathbb{O}_e(\mathcal{M}_i, \pi_j, \epsilon/2, \log(N^2/\delta))$ to evaluate the policy $\pi_j$ on the MDP $\mathcal{M}_i$
9:      Call Subroutine 4 to find a set $\hat{\Pi}$ that covers $(1 - 3\delta)$-fraction of the MDPs in $\hat{\Omega}$
10:      **if** $\sqrt{\frac{|\hat{\Pi}| \log(2N/\delta)}{N - |\hat{\Pi}|}} \leq \delta$ **then**
11:          **Output**: the policy-value set $\hat{\Pi} = \{(\pi_j, v_{j,j}), \forall j \in \mathcal{U}\}$
12:      Double the number of the sampled MDP, i.e. $N = 2N$

**Test Stage**

1: **Input**: the policy-value set $\hat{\Pi}$ from the Pre-train Stage, Episode number $K$
2: Initialize: the MDP set $\hat{\Pi}_1 = \hat{\Pi}$, the phase counter $l = 1, k_0 = 1, \delta = \epsilon = 1/\sqrt{K}$
3: **for** episode $k = 1, \cdots, K$ **do**
4:      Calculate $(\pi_l, v_l) = \arg\max_{(\pi_l, v_l) \in \hat{\Pi}_l} v_l$
5:      Execute the optimal policy $\pi_l$, and receive the total reward $G_k$
6:      **if** $\left| \frac{1}{k - k_0 + 1} \sum_{\tau = k_0}^k G_k - v_l \right| \geq 4\epsilon + \sqrt{\frac{2 \log(4K/\delta)}{k - k_0 + 1}}$ **then**
7:          Eliminate $(\pi_l, v_l)$ from the instance set $\hat{\Pi}_l$, denote the remaining set as $\hat{\Pi}_{l+1}$
8:          Set $k_0 = k + 1$, and $l = l + 1$

---

To tackle the above difficulties, we formulate this problem as a policy candidate collection-elimination process. Our intuition is to find a minimum policy set that can generalize to most

MDPs sampled from $\mathbb{D}$. In the pre-training stage, we maintain a policy set that can perform well on most MDP instances. This includes policies that are the near-optimal policy for an MDP $\mathcal{M}$ with relatively large $\mathcal{P}(\mathcal{M})$, or that can work well on different MDPs. In the test stage, we sequentially execute policies in this set. Once we realize that current policy is not near-optimal for $\mathcal{M}^*$, we eliminate it and switch to another. This helps reduce the regret from the cardinality of the whole state-action space to the size of policy covering set. The pseudo code is in Algorithm 1.

**Pre-training Stage.** In the pre-training stage, we say a policy-value pair $(\pi, v)$ that covers an MDP $\mathcal{M}$ if $\pi$ is $\mathcal{O}(\epsilon)$-optimal for the MDP $\mathcal{M}$ and $v$ is an estimation of the optimal value $V_\mathcal{M}^*(s_1)$ with at most $\mathcal{O}(\epsilon)$ error. For a policy-value set $\hat{\Pi}$, we say the policy set covers the distribution $\mathbb{D}$ with probability at least $1 - \mathcal{O}(\delta)$ if $\mathrm{Pr}_{\mathcal{M} \sim \mathbb{D}}\left[\exists (\pi, v) \in \hat{\Pi}, (\pi, v) \text{ covers } \mathcal{M}\right] \geq 1 - \mathcal{O}(\delta)$. The basic goal in the pre-training phase is to find a policy-value set $\hat{\Pi}$ with bounded cardinality that covers $\mathbb{D}$ with high probability. The pre-training stage contains several phases. In each phase, we sample $N$ MDPs from the distribution $\mathbb{D}$ and obtain an MDP set $\{\mathcal{M}_1, \mathcal{M}_2, \cdots, \mathcal{M}_N\}$. We call the oracle $\mathbb{O}_l$ to calculate the near-optimal policy $\pi_j$ for each MDP $\mathcal{M}_j$, and we calculate the value estimation $v_{i,j}$ for each policy $\pi_j$ on MDP $\mathcal{M}_i$ using oracle $\mathbb{O}_e$. We use the following condition to indicate whether the pair $(\pi_j, v_{j,j})$ covers the MDP $\mathcal{M}_i$:

$$\mathrm{Cnd}(v_{i,j}, v_{i,i}, v_{j,j}) = \mathbb{I}\left[|v_{i,j} - v_{i,i}| < \epsilon\right] \cap \mathbb{I}\left[|v_{i,j} - v_{j,j}| < \epsilon\right].$$

The above condition indicates that $\pi_j$ is a near-optimal policy for $\mathcal{M}_i$, and $v_{j,j}$ is an accurate estimation of the value $V_{\mathcal{M}_i,1}^{\pi_j}(s_1)$. With this condition, we construct a policy-value set $\hat{\Pi}$ that covers $(1 - 3\delta)$-fraction of the MDPs in the sampled MDP set by calling Subroutine 4. We output the policy-value set once the distribution estimation error $\sqrt{\frac{|\hat{\Omega}| \log(2N/\delta)}{N - |\hat{\Omega}|}}$ is less than $\delta$, and double the number of the sampled MDPs $N$ to increase the accuracy of the distribution estimation otherwise. After the pre-training phase, we can guarantee that the returned policy set can cover $\mathbb{D}$ with probability at least $1 - \mathcal{O}(\delta)$, i.e.

$$\mathrm{Pr}_{\mathcal{M} \sim \mathbb{D}}\left[\exists (\pi, v) \in \hat{\Pi}, \left|V_{\mathcal{M},1}^\pi(s_1) - V_{\mathcal{M},1}^*(s_1)\right| < 2\epsilon \cap \left|V_{\mathcal{M},1}^\pi(s_1) - v\right| < 2\epsilon\right] \geq 1 - \mathcal{O}(\delta).$$

**Fine-tuning Stage.** We start with the policy-value set $\hat{\Pi}$ from the pre-training stage and eliminate the policy-value pairs until we reach a $(\pi, v) \in \hat{\Pi}$ that covers the test MDP $\mathcal{M}^*$. Specifically, we split all episodes into different phases. In phase $l$, we maintain a set $\hat{\Pi}_l$ that covers the real environment $\mathcal{M}^*$ with high probability. We select $(\pi_l, v_l)$ with the most optimistic value $v_l$ in $\hat{\Pi}_l$ and execute the policy $\pi_l$ for several episodes. During execution, we also evaluate the policy $\pi_l$ on the MDP $\mathcal{M}^*$ (i.e. $V_{\mathcal{M}^*,1}^{\pi_l}(s_1)$) and maintain the empirical estimation $\frac{1}{k-k_0+1}\sum_{\tau=k_0}^{k} G_k$. Once we identify that $\pi_l$ is not near-optimal for $\mathcal{M}^*$, we end this phase and eliminate $(\pi_l, v_l)$ from $\hat{\Pi}_l$.

### 4.2.2 REGRET

The efficiency of Algorithm 1 is summarized in Theorem 2, and the proof is in Appendix C.

**Theorem 2.** *The regret of Algorithm 1 is at most*

$$\mathrm{Reg}_K(\mathbb{D}, \textit{Alg. 1}) \leq \mathcal{O}\left(\sqrt{\mathcal{C}(\mathbb{D})K\log^2(K)} + \mathcal{C}(\mathbb{D})\right),$$

*where $\mathcal{C}(\mathbb{D}) \triangleq \min_{\mathcal{P}(\tilde{\Omega}) \geq 1-\delta} |\tilde{\Omega}|$ is a complexity measure of $\mathbb{D}$ and $\delta = 1/\sqrt{K}$. In addition, with probability at least $1 - \mathcal{O}\left(\delta \log(\mathcal{C}(\mathbb{D})/\delta)\right)$, samples required in the pre-training stage is $\mathcal{O}(\frac{\mathcal{C}(\mathbb{D})}{\delta^2})$.*

Different from the previous regret bound when the pre-training is unavailable (Azar et al., 2017; Osband et al., 2013; Zhang et al., 2021), this distribution-dependent upper bound improves the dependence on $S, A$ to a complexity measure of $\mathbb{D}$ defined as $\mathcal{C}(\mathbb{D})$. $\mathcal{C}(\cdot)$ serves as a multiplied factor, and can be small when $\mathbb{D}$ enjoys benign properties as shown below.

First, when the cardinality of $\Omega$ is small, i.e. $|\Omega| \ll SA$, we have $\mathcal{C}(\mathbb{D}) \leq |\Omega|$ and the pre-training can greatly reduce the instance space via representation learning or multi-task learning (Agarwal et al., 2020; Brunskill & Li, 2013). Specifically, the regret in the test stage is reduced from

$\tilde{\mathcal{O}}(\sqrt{SAHK})$ to $\tilde{\mathcal{O}}(\sqrt{|\Omega|K})$. To the best of our knowledge, this is the first result that achieves dependence only on $M$ for any general distribution in RL generalization. We provide a lower bound in Appendix C.4 to show that our regret in the test stage is near-optimal except for logarithmic factors.

When $|\Omega|$ is large or even infinite, $\mathcal{C}(\mathbb{D})$ is still bounded and can be significantly smaller when $\mathbb{D}$ enjoys benign properties. In fact, in the worst case, it is not hard to find that the dependence on $\mathcal{C}(\mathbb{D})$ is unavoidable[2], and there is no way to expect any improvement. However, in practice where $|\Omega|$ is large, the probability will typically be concentrated on several subset region of $\Omega$ and decay quickly outside, e.g. when $\mathbb{D}$ is subgaussian or mixtures of subgaussian. Specifically, if the probability of the $i$-th MDP $p_i \le c_1 e^{-\lambda i}$ for some positive constant $c_1, \lambda$, then $\mathcal{C}(\mathbb{D}) \le \mathcal{O}(\log \frac{1}{\delta})$, which gives the

upper bound $\mathcal{O}\left(\sqrt{K \log^3(K)}\right)$.

It is worthwhile to mention that our algorithm in the pre-training stage actually finds a *policy* covering set, rather than an MDP covering set, despite that the regret in Theorem 2 depends on the cardinality of the MDP covering set. This dependence can possibly be improved to the cardinality of the policy covering set by adding other assumptions to our policy learning oracle, which we leave as an interesting problem for the future research.

## 5 RESULTS FOR THE SETTING WITHOUT TEST-TIME INTERACTION

In this section, we study the benefits of pre-training without test-time interaction. As illustrated by Proposition 1, we only pursue the optimality *in expectation*, which is in line with supervised learning. Traditional Empirical Risk Minimization algorithm can be $\epsilon$-optimal with $\mathcal{O}(\frac{\log |\mathcal{H}|}{\epsilon^2})$ samples, and we expect this to be true in RL generalization as well. However, a policy in RL needs to sequentially interact with the environment, which is not captured by pure ERM algorithm. In addition, different MDPs in $\Omega$ can have distinct optimal actions, making it hard to determine which action is better even in expectation. To overcome these issues, we design an algorithm called OMERM in Algorithm 2. Our algorithm is designed for tabular MDPs with finite state-actions space. Nevertheless, it can be extended to the non-tabular setting by combining our ideas with previous algorithms for efficient RL with function approximation (Jin et al., 2020a; Wang et al., 2020).

In Algorithm 2, we first sample $N$ tasks from the distribution $\mathbb{D}$ as the input. The goal of this algorithm is to find a near-optimal policy in expectation w.r.t. the task set $\{\mathcal{M}_1, \mathcal{M}_2, \cdots, \mathcal{M}_N\}$. In each episode, the algorithm estimates each MDP instance using history trajectories, and calculates the optimistic value function of each instance. Based on this, it selects the policy from $\Pi$ that maximizes the average optimistic value, i.e. $\frac{1}{N}\sum_{i=1}^{N} \hat{V}_{\mathcal{M}_i,1}^{\pi}(s_1)$. This selection objective is inspired by ERM, with the difference that we require this estimation to be optimistic. It can be achieved by a planning oracle when $\Pi$ is all stochastic maps from $(\mathcal{S}, H)$ to $\Delta(\mathcal{A})$, or by gradient descent when $\Pi$ is a parameterized model. For each sampled $\mathcal{M}_i$, our algorithm needs to interact with it for $\text{poly}(S, A, H, \frac{1}{\epsilon})$ times, which is a typical requirement to learn a model well.

**Theorem 3.** *With probability[3] at least $2/3$, Algorithm 2 can output a policy $\hat{\pi}$ satisfying* $\mathbb{E}_{\mathcal{M}^* \sim \mathbb{D}}[V_{\mathcal{M}^*}^{\pi^*(\mathbb{D})} - V_{\mathcal{M}^*}^{\hat{\pi}}] \le \epsilon$ *with* $\mathcal{O}\left(\frac{\log \mathcal{N}_{\epsilon/(12H)}^{\Pi}}{\epsilon^2}\right)$ *MDP instance samples during training. The number of episodes collected for each task is bounded by* $\mathcal{O}\left(\frac{H^2 S^2 A \log(SAH)}{\epsilon^2}\right)$.

We defer the proof of Theorem 3 to Appendix D. This theorem implies that Algorithm 2 needs approximately $\mathcal{O}(\log(\mathcal{N}_{\frac{\epsilon}{12H}}^{\Pi})/\epsilon^2)$ samples to return an $\epsilon$-optimal policy *in expectation*. Recall that $\log \mathcal{N}_{\frac{\epsilon}{12H}}^{\Pi}$ is the log-covering number of $\Pi$. When $\Pi$ is all stochastic maps from $\mathcal{S}, H$ to $\Delta(\mathcal{A})$, it is bounded by $\tilde{\mathcal{O}}(HSA)$. When $\Pi$ is a parameterized model where the parameter $\theta \in \mathbb{R}^d$ has finite norm and $\pi_\theta$ satisfies some smoothness condition on $\theta$, $\log(\mathcal{N}_{\frac{\epsilon}{12H}}^{\Pi}) \le \tilde{\mathcal{O}}(d)$. This result

---

[2]Consider the bandit case where there are $M$ arms. The optimal arm is arm $i$ in $\mathcal{M}_i$, and $\mathbb{D}$ is a uniform distribution over $M$ MDPs. $\mathcal{C}(\mathbb{D}) = M$ in this case, and the $M$ dependence is unavoidable in this hard instance since the agent has to independently explore and test whether each arm is the optimal arm.

[3]This probability can be further improved to $1 - \delta$ by executing Algorithm 2 for $\log(1/\delta)$ times and then returning a policy with maximum average value. Please see Appendix D.2 for the detailed discussion.

---

**Algorithm 2** OMERM (**O**ptimistic **M**odel-based **E**mpirical **R**isk **M**inimization)

**Input**: target accuracy $\epsilon > 0$

**Input**: the sampled $N$ MDPs denoted as $\{\mathcal{M}_1, \mathcal{M}_2, \cdots, \mathcal{M}_N\}$ from distribution $\mathbb{D}$, with $N = C_1 \log\left(\mathcal{N}\left(\Pi, \epsilon/(12H), d\right)\right)/\epsilon^2$ for a constant $C_1 > 0$

$K = C_2 S^2 A H^2 \log(SAH/\epsilon)/\epsilon^2$ for constants $C_2 > 0$

**for** episode $k = 1, 2, \cdots, K$ **do**

5:      **for** $i = 1, 2, \cdots, N$ **do**

Denote $N_{\mathcal{M}_i, k, h}(s, a, s')$ and $N_{\mathcal{M}_i, k, h}(s, a)$ as the counter that agent encounters $(s, a, s')$ and $(s, a)$ at step $h$ in $\mathcal{M}_i$ till step $k - 1$, respectively

Estimate $\hat{\mathbb{P}}_{\mathcal{M}_i, k, h}(s'|s, a) = \frac{N_{\mathcal{M}_i, k, h}(s, a, s')}{\max\{1, N_{\mathcal{M}_i, k, h}(s, a)\}}$ for step $h \in [H]$

Estimate $\hat{R}_{\mathcal{M}_i, k, h}(s, a) = \frac{\sum_{\tau=1}^{k-1} r_{\mathcal{M}_i, \tau, h} \mathbb{1}(s_{\mathcal{M}_i, \tau, h} = s, a_{\mathcal{M}_i, \tau, h} = a)}{\max\{1, N_{\mathcal{M}_i, k, h}(s, a)\}}$ for step $h \in [H]$

Define the UCB bonus $b_{\mathcal{M}_i, k, h}(s, a) = \sqrt{\frac{8S \log(8SANHK)}{\max\{1, N_{\mathcal{M}_i, k, h}(s, a)\}}}$

10:      Initialize $\hat{V}^{\pi}_{\mathcal{M}_i, k, H+1}(s, a) = 0, \forall s, a$

**for** $h = H, H-1, \cdots, 1$ **do**

$\hat{Q}^{\pi}_{\mathcal{M}_i, k, h}(s, a) = \min\left\{1, \hat{R}_{\mathcal{M}_i, k, h}(s, a) + b_{\mathcal{M}_i, k}(s, a) + \hat{\mathbb{P}}_{\mathcal{M}_i, k, h} \hat{V}^{\pi}_{\mathcal{M}_i, k, h+1}(s, a)\right\}$

$\hat{V}^{\pi}_{\mathcal{M}_i, k, h}(s) = \sum_a \pi_h(a|s) \hat{Q}^{\pi}_{\mathcal{M}_i, k, h}(s, a)$

Calculate the optimistic policy $\pi_k = \arg\max_{\pi \in \Pi} \frac{1}{N} \sum_{i=1}^{N} \hat{V}^{\pi}_{\mathcal{M}_i, 1}(s_1)$.

15:      **for** $i = 1, 2, \cdots, N$ **do**

Execute the policy $\pi_k$ on MDP $\mathcal{M}_i$ for one episode, and observe the trajectory $(s_{\mathcal{M}_i, k, h}, a_{\mathcal{M}_i, k, h}, r_{\mathcal{M}_i, k, h})_{h=1}^{H}$.

**Output**: a policy selected uniformly randomly from the policy set $\{\pi_k\}_{k=1}^{K}$.

---

matches traditional bounds in supervised learning, and it implies that when we pursue the optimal *in expectation*, generalization in RL enjoys quantitatively similar upper bound to supervised learning.

## 6    CONCLUSION AND FUTURE WORK

This work theoretically studies how much pre-training can improve test performance under different generalization settings. we first point out that RL generalization is fundamentally different from the generalization in supervised learning, and fine-tuning on the target environment is necessary for good generalization. When the agent can interact with the test environment to update the policy, we first prove that the prior information obtained in the pre-training stage can be theoretically useless in the asymptotic setting, and show that in non-asymptotic setting we can reduce the test-time regret to $\tilde{O}\left(\sqrt{C(\mathbb{D})K}\right)$ by designing an efficient learning algorithm. In addition, when the agent cannot interact with the test environment, we provide an efficient algorithm called OMERM which returns a near-optimal policy *in expectation* by interacting with $\mathcal{O}\left(\log(\mathcal{N}^{\Pi}_{\epsilon/(12H)})/\epsilon^2\right)$ MDP instances.

Our work seeks a comprehensive understanding on how much pre-training can be helpful to test performance theoretically, and it also provides insights on real RL generalization application. For example, when test time interactions are not allowed, one cannot guarantee to be near optimal *in instance*. Therefore, for a task where large regret is not tolerable, instead of designing good algorithm, it is more important to find an environment that is close to the target environment and improve policies there, rather than to train a policy in a diverse set of MDPs and hope it can generalize to the target. In addition, for tasks where we can improve policies on the fly, we can try to pre-train our algorithm in advance to reduce the regret suffered. This corresponds to the many applications where test time interactions are very expensive, such as autonomous driving and robotics.

There are still problems remaining open. Firstly, we mainly study the *i.i.d.* case where the training MDPs and test MDPs are sampled from the same distribution. It is an interesting problem to study the out-of-distribution generalization in RL under certain distribution shifting. Secondly, there is still room for improving our instance dependent bound possibly by leveraging ideas from recent Bayesian-optimal algorithms such as Thompson sampling (Osband et al., 2013). We hope these problems can be addressed in the future research.

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

## A  OMITTED PROOF FOR PROPOSITION 1

Assume $\Omega$ is the set of $M$ MDP instances $\mathcal{M}_1, \cdots, \mathcal{M}_M$, where all instances consist only one identical state $s_1$ and their horizon $H = 1$. In $\mathcal{M}_i$ and state $s_1$, the reward is 1 for action $a_i$ and 0 otherwise. The optimal policy in $\mathcal{M}_i$ is therefore taking action $a_t$ and $V_{\mathcal{M}_i}^{\pi^*(\mathcal{M}_i)} = 1$.

For any distribution $\mathbb{D}$, assume the probability to sample $\mathcal{M}_i$ is $p_i > 0$. For any deployed policy $\hat{\pi}$, assume the probability that $\hat{\pi}$ takes action $a_i$ is $q_i$, then

$$\mathbb{E}_{\mathcal{M}^* \sim \mathbb{D}} V(\mathcal{M}^*, \hat{\pi}) = \sum_{i=1}^{M} p_i q_i \leq \max_{i \in [M]} p_i.$$

Therefore, denote $\epsilon_0 = 1 - \max_{i \in [M]} p_i > 0$, we have

$$\mathbb{E}_{\mathcal{M}^* \sim \mathbb{D}} \left[ V_{\mathcal{M}^*}^{\pi^*(\mathcal{M}^*)} - V_{\mathcal{M}^*}^{\hat{\pi}} \right] \geq \epsilon_0.$$

Notice that when $\mathbb{D}$ is a uniform distribution, $\epsilon_0 = \frac{M-1}{M}$, which can be arbitrarily close to 1.

## B  OMITTED PROOF FOR THEOREM 1

To prove the theorem, we let $\Omega$ be a subset of MDP with $H = S = 1$, under which it becomes a bandit problem, and it suffices to prove the theorem in this setting. Below we first introduce the bandits notations, and give the complete proof.

### B.1  NOTATIONS

To be consistent with traditional $K$-arm bandits problem, in this section we use a slightly different notations from the main paper. An bandit instance can be represented as a vector $r \in \Omega \subset [0,1]^K$, where $r_k$ is the mean reward when arm $k$ is pulled. Inherited from previous MDP settings, we also assume that this reward is sampled from an 1-subgaussian distribution with mean $r_k$. In each episode $t \in [T]$, based on initial input and the history, an algorithm $\mathcal{A}$ chooses an arm $a_t$ to pull, and obtain a reward $y_{a_t} \sim \mathbb{D}_{a_t}$. Similarly, assume $r$ is sampled from a distribution $\mathbb{D}$ supported on $\Omega$, and we want to minimize the Bayesian regret

$$\text{Reg}_T(\mathbb{D}, \mathcal{A}) \triangleq \mathbb{E}_{r \in \mathbb{D}} \text{Reg}_T(r, \mathcal{A}), \text{ where } \text{Reg}_T(r, \mathcal{A}) \triangleq \mathbb{E}_{\mathcal{A}} \sum_{t=1}^{T} [r^* - r_{a_t}].$$

Here $r^* = \max_k r_k$ is the optimal arm. If we define $S_k^T(r, \mathcal{A}) = \sum_{t=1}^{T} \mathbb{I}[a_t = k]$ as the r.v. of how many times $\mathcal{A}$ has pulled arm $k$ in $T$ episodes and $\Delta_k = r^* - r_k$ as the sub-optimal gap, we can decompose regret

$$\text{Reg}_T(r, \mathcal{A}) = \sum_{k=1}^{K} \Delta_k \mathbb{E}[S_k^T(r, \mathcal{A})]. \tag{1}$$

This identity is frequently used in the subsequent proof.

## B.2 PROOF

We first specify the choice of support, constant and algorithm. Without loss of generality, we set $\Omega = [0,1]^K$, which is quite common and general in bandit tasks. Let $c_0 = \frac{1}{16}$, and $\hat{\mathcal{A}}$ be the Asymptotically Optimal UCB Algorithm defined in Algorithm 3. On the other hand, let $\tilde{\mathcal{A}}$ be uniformly optimal, i.e. $\tilde{\mathcal{A}}(\mathbb{D},T) = \arg\min_{\mathcal{A}} \operatorname{Reg}_T(\mathbb{D}, \mathcal{A})$. To prove the theorem, we only need to show that

$$\liminf_{T \to \infty} \frac{\operatorname{Reg}_T(\mathbb{D}, \tilde{\mathcal{A}}(\mathbb{D},T))}{\operatorname{Reg}_T(\mathbb{D}, \hat{\mathcal{A}}(T))} \geq c_0, \tag{2}$$

which is the major result (point (2)). Later, we show that $\lim_{T \to \infty} \operatorname{Reg}_T(\mathbb{D}, \tilde{\mathcal{A}}(\mathbb{D},T)) = +\infty$ in Lemma 9, which proves point (1) in the theorem. When $T$ is fixed, we abbreviate $\tilde{\mathcal{A}}(\mathbb{D},T), \hat{\mathcal{A}}(T)$ as $\tilde{\mathcal{A}}, \hat{\mathcal{A}}$.

$\hat{\mathcal{A}}$ enjoys a well known instance dependent regret upper bound, which is copied below:

**Lemma 4** (Theorem 8.1 in Lattimore & Szepesvári (2020) ). *For all $r \in \Omega, \forall k \in [K]$,*

$$\mathbb{E}[S_k^T(r, \hat{\mathcal{A}})] \leq \min\{T, \inf_{\varepsilon \in (0, \Delta_k)} \left(1 + \frac{5}{\varepsilon^2} + \frac{2\big(\log(T \log^2 T + 1) + \sqrt{\pi \log(T \log^2 T + 1)} + 1\big)}{(\Delta_k - \varepsilon)^2}\right)\}.$$

By Holder Inequality, we immediately have

$$\Delta_k \mathbb{E}[S_k^T(r, \hat{\mathcal{A}})] \leq \min\{\Delta_k T, \Delta_k + \frac{u(T) \log T}{\Delta_k}\},$$

where the coefficient function

$$u(T) = \sup_{t \geq T} \frac{1}{\log t} \left[5^{1/3} + (2 \log(t \log^2 t + 1) + \sqrt{\pi \log(t \log^2 t + 1)} + 1)^{1/3}\right]^3$$

being non-increasing and $\lim_{T \to \infty} u(T) = 2$. For simplicity, we define $e(T) \triangleq \sqrt{\frac{u(T) \log T}{T-1}}$ and $s(\Delta, T) \triangleq \min\{\Delta T, \Delta + \frac{u(T)}{\Delta} \log T\}$. We can upper bound $\operatorname{Reg}_T(r, \hat{\mathcal{A}}) \leq \sum_{k=1}^{K} s(\Delta_k, T)$.

**Decomposition of Inq. 2** For $k \in [K]$, define $\Omega_T^k = \{r \in \Omega : \Delta_k \geq T^{-p_0}\}$, where $p_0 \in (0, \frac{1}{2})$ is a universal constant to be specified later. Further define $\Lambda_k^\epsilon = \{r \in \Omega, r_k + \Delta_k(1 + \epsilon) < 1\}$. Using Eq. 1, it suffices to prove the lemma if $\forall k \in [K]$,

$$\liminf_{T \to \infty} \frac{\int_\Omega p(r) \Delta_k \mathbb{E}[S_k^T(r, \tilde{\mathcal{A}})] \mathrm{d}r}{\int_\Omega p(r) \Delta_k \mathbb{E}[S_k^T(r, \hat{\mathcal{A}})] \mathrm{d}r} \geq c_0. \tag{3}$$

Fix $k \in [K]$ and $\epsilon > 0$ that is sufficiently small, we decompose Inq. 3 as below:

$$\frac{\int_\Omega p(r) \Delta_k \mathbb{E}[S_k^T(r, \tilde{\mathcal{A}})] \mathrm{d}r}{\int_\Omega p(r) \Delta_k \mathbb{E}[S_k^T(r, \hat{\mathcal{A}})] \mathrm{d}r} \geq \frac{\int_{\Omega_T^k \cap \Lambda_k^\epsilon} p(r) \Delta_k \mathbb{E}[S_k^T(r, \tilde{\mathcal{A}})] \mathrm{d}r}{\int_{\Omega_T^k \cap \Lambda_k^\epsilon} p(r) s(\Delta_k, T) \mathrm{d}r} \cdot \frac{\int_{\Omega_T^k \cap \Lambda_k^\epsilon} p(r) s(\Delta_k, T) \mathrm{d}r}{\int_{\Omega_T^k} p(r) s(\Delta_k, T) \mathrm{d}r}$$

$$\times \frac{\int_{\Omega_T^k} p(r) s(\Delta_k, T) \mathrm{d}r}{\int_\Omega p(r) s(\Delta_k, T) \mathrm{d}r} \cdot \frac{\int_\Omega p(r) s(\Delta_k, T) \mathrm{d}r}{\int_\Omega p(r) \Delta_k \mathbb{E}[S_k^T(r, \hat{\mathcal{A}})] \mathrm{d}r}$$

Sequentially denote the four terms in the right hand side as $M_1 \sim M_4$. $M_4$ can be lower bounded by 1 based on the Lemma 4. The rest three terms is bounded by the subsequent three lemmas.

**Lemma 5.** $\forall k \in [K]$,

$$\liminf_{T \to \infty} \frac{\int_{\Omega_T^k} p(r) s(\Delta_k, T) \mathrm{d}r}{\int_\Omega p(r) s(\Delta_k, T) \mathrm{d}r} \geq 2p_0. \tag{4}$$

Lemma 5 lower bounds $M_3$, saying that the influence of instance in $\Omega \setminus \Omega_T^k$ is negligible. The proof of this lemma needs theory on Lebesgue Integral and Radon Transform, which will be introduced in the Section B.3.

**Lemma 6.** *For any $k \in [K], \epsilon \in (0,1)$,*

$$\lim_{T \to \infty} \frac{\int_{\Omega_T^k \bigcap \Lambda_k^\epsilon} p(r)s(\Delta_k, T)\mathrm{d}r}{\int_{\Omega_T^k} p(r)s(\Delta_k, T)\mathrm{d}r} = 1.$$

Lemma 6 implies that $M_2 \to 1$, allowing us to focus on the integral on a calibrated smaller set $\Omega_T^k \bigcap \Lambda_k^\epsilon$, in which we can use information theory to control $\mathbb{E}[S_k^T(r, \tilde{\mathcal{A}})]$. The following lemma is the major lemma in our proof, which use the optimality of $\tilde{\mathcal{A}}$ to analyze the global structure of $\mathrm{Reg}_T(r, \tilde{\mathcal{A}})$ for $r \in \Omega_T^k \bigcap \Lambda_k^\epsilon$, and lower bound term $M_1$:

**Lemma 7.** *For any $\epsilon \in (0,1)$, we have*

$$\liminf_{T \to \infty} \frac{\int_{\Omega_T^k \bigcap \Lambda_k^\epsilon} p(r)\Delta_k \mathbb{E}[S_k^T(r, \tilde{\mathcal{A}})]\mathrm{d}r}{\int_{\Omega_T^k \bigcap \Lambda_k^\epsilon} p(r)s(\Delta_k, T)\mathrm{d}r} \geq \frac{\frac{1}{2} - 2p_0}{(1+\epsilon)^2}.$$

Combined together,

$$\liminf_{T \to \infty} \frac{\int_\Omega p(r)\Delta_k \mathbb{E}[S_k^T(r, \tilde{\mathcal{A}})]\mathrm{d}r}{\int_\Omega p(r)\Delta_k \mathbb{E}[S_k^T(r, \hat{\mathcal{A}})]\mathrm{d}r} \geq \frac{2p_0(\frac{1}{2} - 2p_0)c_k}{(1+\epsilon)^2}, \tag{5}$$

The proof of Inq. 2 is finished by selecting $p_0 = \frac{1}{8}$ and letting $\epsilon \to 0$.

---

**Algorithm 3** $\hat{\mathcal{A}}$: Asymptotically UCB

---
1: Input $\Omega = [0,1]^K$, total episode $T$
2: **for** step $t = 1, \cdots, K$ **do**
3:     Choose arm $a_t$, and obtain reward $y_t$
4:     Set $\hat{R}_t = y_t, \hat{S}_t = 1$
5: **for** step $t = K+1, \cdots, T$ **do**
6:     $f(t) = 1 + t\log^2(t)$
7:     Choose $a_t = \arg\max_k \left( \frac{\hat{R}_k}{\hat{S}_k} + \sqrt{\frac{2\log f(t)}{\hat{S}_k}} \right)$
8:     Set $\hat{R}_k = R_k + y_t, S_k = S_k + 1$

---

### B.3 PROOF OF LEMMAS

Before proving all lemmas above, we need to introduce theory on Lebesgue Integral and Radon Transform. In $\mathbb{R}^K$ space where $K \geq 2$, when a function is Riemann integrable, it is also Lebesgue integrable, and two integrals are equal. Since we assume $p(r) \in C(\Omega)$, the integral always exists. Below we always consider the Lebesgue integral. For a compact measurable set $S$, define $\mathcal{L}^p(S)$ as the space of all measurable function in $S$ with standard $p$-norm. Since the p.d.f of $\mathbb{D}$ is continuous in compact set $\Omega$ and is positive, $\exists L, U \in \mathbb{R}^+, \forall r \in \Omega, p(r) \in [L, U]$.

Denote $\mathcal{T}_k = \{r \in \Omega : r_k = \max(r)\}$. Clearly, $\int_\Omega f(r)\mathrm{d}r = \sum_{k \in [K]} \int_{\mathcal{T}_k} f(r)\mathrm{d}r$ since $m(\mathcal{T}_i \bigcap \mathcal{T}_j) = 0, \forall \mathcal{T}_i \neq \mathcal{T}_j$. Here $m(\cdot)$ is the Lebesgue measure in $\mathbb{R}^K$. If we define $\mathcal{P}_{t,\gamma} = \{r \in \Omega, \gamma \cdot r = t\}$ and $\gamma_k^i = (e_i - e_k), i \neq k$, then $\mathcal{P}_{t,\gamma_k^i} \bigcap \mathcal{T}_i = \{r \in \mathcal{T}_i : \Delta_k = t\}$. According to Radon Transform theory, since $\mathcal{T}_i$ is compact and $p(r) \in C(\mathcal{T}_i)$,

$$\rho_k^i(t) \triangleq \int_{\mathcal{P}_{t,\gamma_k^i} \bigcap \mathcal{T}_i} p(r)\mathrm{d}r,$$

is also continuous w.r.t variable $t \in [0,1]$ for all $k \neq i \in [K]$. Here the integration is perform in the corresponding $\mathbb{R}_{K-1}$ space, i.e. the plane $\mathcal{P}_{t,\gamma_k^i}$, and when $m_{K-1}(\mathcal{P}_{t,\gamma_k^i} \bigcap \mathcal{T}_i) > 0$, $\frac{\rho_k^i(t)}{m_{K-1}(\mathcal{P}_{t,\gamma_k^i} \bigcap \mathcal{T}_i)} \in [L, U]$.

We further define $q_k(t) = \sum_{i \neq k} \rho_k^i(t)$, which also belongs to $C([0,1])$. The continuity of $q_k(t)$ helps derive the following equation. For any $f \in L^1(\Omega)$ that relies only on $\Delta_k$, i.e. $f(r) = \tilde{f}(\Delta_k)$, we have

$$
\begin{aligned}
\int_\Omega p(r)f(r)\mathrm{d}r &= \sum_{i \in [K]} \int_{\mathcal{T}_i} p(r)f(r)\mathrm{d}r \\
&= \sum_{i \in [K]} \int_{\mathcal{T}_i} p(r)\tilde{f}(r_i - r_k)\mathrm{d}r \\
&= \frac{\sqrt{2}}{2} \sum_{i \in [K]} \int_{[0,1]} \tilde{f}(t)\rho_k^i(t)\mathrm{d}\Delta = \frac{\sqrt{2}}{2} \int_{[0,1]} \tilde{f}(t)q_k(t)\mathrm{d}\Delta.
\end{aligned}
\tag{6}
$$

Here the last line is because of Fubini Theorem, which states that the integral of a function can be computed by iterating lower-dimensional integrals in any order. The factor $\frac{\sqrt{2}}{2}$ arises because in traditional Radon Transform we require $\|\gamma\|_2 = 1$, but here $\|\gamma_k^i\|_2 = \|e_i - e_k\|_2 = \sqrt{2}$.

### B.3.1 Proof of Lemma 5

Recall that $s(\Delta, T) = \min\{\Delta T, \Delta + \frac{u(T)}{\Delta} \log T\}$, and $\Delta T < \Delta + \frac{u(T)}{\Delta} \log T \leftrightarrow \Delta < e(T)$, where $e(T)$ is defined as $\sqrt{\frac{u(T) \log T}{T - 1}}$. When $e(T) \leq T^{-p_0}$, Eq. 6 implies that $\forall r \in \Omega_T^k$,

$$
\begin{aligned}
\liminf_{T \to \infty} \frac{\int_{\Omega_T^k} p(r)s(\Delta_k, T)\mathrm{d}r}{\int_\Omega p(r)s(\Delta_k, T)\mathrm{d}r} &= \liminf_{T \to \infty} \frac{\int_\Omega p(r)s(\Delta_k, T)\mathbb{I}[\Delta_k \geq T^{-p_0}]\mathrm{d}r}{\int_\Omega p(r)s(\Delta_k, T)\mathrm{d}r} \\
&= \liminf_{T \to \infty} \frac{\int_{[0,1]} q_k(\Delta)s(\Delta, T)\mathbb{I}[\Delta \geq T^{-p_0}]\mathrm{d}\Delta}{\int_{[0,1]} q_k(\Delta)s(\Delta, T)\mathrm{d}\Delta}.
\end{aligned}
$$

To prove the lemma, it suffices to show that

$$
\limsup_{T \to \infty} \frac{\int_{[0,1]} q_k(t)s(\Delta, T)\mathbb{I}[\Delta \leq T^{-p_0}]\mathrm{d}\Delta}{\int_{[0,1]} q_k(\Delta)s(\Delta, T)\mathbb{I}[\Delta \geq T^{-p_0}]\mathrm{d}\Delta} \leq \frac{0.5 - p_0}{p_0}.
\tag{7}
$$

Define $E_T = \{x \in [0,1] : x \leq e(T)\}$, $F_T = \{x \in [0,1] : x \leq T^{-p_0}\}$, then

$$
\lim_{T \to \infty} \frac{\int_{E_n} q_k(\Delta)s(\Delta, T)\mathrm{d}\Delta}{\int_{F_n \backslash E_n} q_k(\Delta)s(\Delta, T)\mathrm{d}\Delta} = \lim_{T \to \infty} \frac{\int_{E_n} q_k(\Delta)\Delta T \mathrm{d}\Delta}{\int_{F_n \backslash E_n} q_k(\Delta)(\Delta + \frac{u(T) \log T}{\Delta})\mathrm{d}\Delta}
\tag{8}
$$

$$
\leq \lim_{T \to \infty} \frac{1}{e^2(T)} \frac{\int_{E_n} q_k(\Delta)\Delta \mathrm{d}\Delta}{\int_{F_n \backslash E_n} q_k(\Delta)\frac{1}{\Delta}\mathrm{d}\Delta}
\tag{9}
$$

We have shown that $\frac{\rho_k^i(t)}{m_{K-1}(\mathcal{P}_{t,\gamma_k^i})} \in [L, U]$. With some calculation, $m(t) \triangleq m_{K-1}(\mathcal{P}_{t,\gamma_k^i} \bigcap \mathcal{T}_i) = \frac{1 - t^{K-1}}{K-1}$. Therefore, $q_k(t) \in [m(t)(K-1)L, m(t)(K-1)U]$, and $q_k(0) = \lim_{t \to 0^+} q_k(t) \geq L > 0$. By this continuity, for small enough $\varepsilon_1 > 0, \exists \delta_1 > 0, \forall t \in [0, \delta_1], q_k(t) \in [(1 - \varepsilon_1)q_k(0), (1 + \varepsilon_1)q_k(0)]$. As a result,

$$
\begin{aligned}
\lim_{T \to \infty} \frac{1}{e^2(T)} \frac{\int_{E_n} q_k(\Delta)\Delta \mathrm{d}\Delta}{\int_{F_n \backslash E_n} q_k(\Delta)\frac{1}{\Delta}\mathrm{d}\Delta} &\leq \lim_{T \to \infty} \frac{1 - \varepsilon_1}{e^2(T)(1 + \varepsilon_1)} \frac{\int_{E_n} \Delta \mathrm{d}\Delta}{\int_{F_n \backslash E_n} \frac{1}{\Delta}\mathrm{d}\Delta} \\
&= \lim_{T \to \infty} \frac{1 - \varepsilon_1}{e^2(T)(1 + \varepsilon_1)} \frac{\frac{1}{2}e^2(T)}{\frac{1}{2}\log \frac{T-1}{u(T)\log T} - p_0 \log T} \\
&= \lim_{T \to \infty} \frac{1 - \varepsilon_1}{2(1 + \varepsilon_1)} \frac{1}{(\frac{1}{2} - p_0)\log T - \frac{1}{2}\log \frac{u(T)T\log T}{T-1}} \\
&= 0.
\end{aligned}
\tag{10}
$$

This implies that the limitation of 8 exists and equals to 0. Similarly, define $G_T = \{x \in [0,1] : x \le \frac{1}{\log T}\}$, then $G_T, F_T \subset [0, \delta_1]$ for sufficiently large $T$. This implies that

$$
\begin{aligned}
\limsup_{T\to\infty} \frac{\int_{[0,1]} q_k(\Delta)s(\Delta,T)\mathbb{I}[\Delta \le T^{-p_0}]\mathrm{d}\Delta}{\int_{[0,1]} q_k(\Delta)s(\Delta,T)\mathbb{I}[\Delta \ge T^{-p_0}]\mathrm{d}\Delta} &= \limsup_{T\to\infty} \frac{\int_{F_n} q_k(\Delta)s(\Delta,T)\mathrm{d}\Delta}{\int_{[0,1]\setminus F_n} q_k(\Delta)s(\Delta,T)\mathrm{d}\Delta} \\
&= \limsup_{T\to\infty} \frac{\int_{F_n\setminus E_n} q_k(\Delta)(\Delta + \frac{u(T)\log T}{\Delta})\mathrm{d}\Delta}{\int_{[0,1]\setminus F_n} q_k(\Delta)(\Delta + \frac{u(T)\log T}{\Delta})\mathrm{d}\Delta} \\
&\le \limsup_{T\to\infty} \frac{\int_{F_n\setminus E_n} q_k(\Delta)\frac{1}{\Delta}\mathrm{d}\Delta}{\int_{G_n\setminus F_n} q_k(\Delta)\frac{1}{\Delta}\mathrm{d}\Delta} \\
&\le \frac{1+\varepsilon_1}{1-\varepsilon_1} \limsup_{T\to\infty} \frac{\int_{F_n\setminus E_n} \frac{1}{\Delta}\mathrm{d}\Delta}{\int_{G_n\setminus F_n} \frac{1}{\Delta}\mathrm{d}\Delta} \\
&= \frac{1+\varepsilon_1}{1-\varepsilon_1} \limsup_{T\to\infty} \frac{\log e(T) - p_0 \log T}{p_0 \log T - \log(\log T)} \\
&= \frac{1+\varepsilon_1}{1-\varepsilon_1} \frac{\frac{1}{2} - p_0}{p_0}.
\end{aligned}
$$

Here the second equation comes from Inq. 10. The proof of Inq. 7 is finished by letting $\varepsilon_1 \to 0$.

### B.3.2 PROOF OF LEMMA 6

Recall that $\Lambda_k^\epsilon$ is defined as $\{r \in \Omega, r_k + \Delta_k(1+\epsilon) < 1\}$. For all $t \in [0,1), i \ne k$,

$$
\begin{aligned}
\mathcal{P}_{t,\gamma_k^i} \bigcap \mathcal{T}_i \bigcap \Lambda_k^\epsilon &= \{r \in \mathcal{P}_{t,\gamma_k^i} : r_i = \max_j r_j, r \in \Lambda_k^\epsilon\} \\
&= \{r \in \Omega : r_i = \max_j r_j, r_i - r_k = t, r \in \Lambda_k^\epsilon\} \\
&= \{r \in \Omega : r_i = \max_j r_j, r_i - r_k = t, r_i + \epsilon t < 1\} \\
&= \{r \in \mathcal{P}_{t,\gamma_k^i} \bigcap \mathcal{T}_i : r_i < 1 - \epsilon t\}.
\end{aligned}
$$

This implies that

$$
m_{K-1}(\mathcal{P}_{t,\gamma_k^i} \bigcap \mathcal{T}_i \setminus \Lambda_k^\epsilon) = m_{K-1}(\{r \in \mathcal{P}_{t,\gamma_k^i} \bigcap \mathcal{T}_i : r_i \in [1-\epsilon t, 1]\}) = \begin{cases} \frac{1-(1-\epsilon t)^{K-1}}{K-1} & t \le \frac{1}{1+\epsilon} \\ \frac{1-t^{K-1}}{K-1} & o.w. \end{cases}.
$$

Notice that $\Lambda_k^\epsilon$ is open, $\mathcal{T}_i \setminus \Lambda_k^\epsilon$ is compact, and $(S_1 \bigcap S_2) \setminus S_3 = S_1 \bigcap (S_2 \setminus S_3)$. We have

$$
\tilde{\rho}_k^i(t) \triangleq \int_{\mathcal{P}_{t,\gamma_k^i} \bigcap (\mathcal{T}_i \setminus \Lambda_k^\epsilon)} p(r)\mathrm{d}r \in C(\mathcal{T}_i \setminus \Lambda_k^\epsilon).
$$

For all $i \ne k$ (Define $O = [T^{-p_0}, 1]$),

$$
\begin{aligned}
\lim_{T\to\infty} \frac{\int_{\Omega_T^k \bigcap \mathcal{T}_i \setminus \Lambda_k^\epsilon} p(r)s(\Delta_k,T)\mathrm{d}r}{\int_{\Omega_T^k \bigcap \mathcal{T}_i} p(r)s(\Delta_k,T)\mathrm{d}r} &= \lim_{T\to\infty} \frac{\int_O s(\Delta,T)\tilde{\rho}_k^i(\Delta)\mathrm{d}\Delta}{\int_O s(\Delta,T)\rho_k^i(\Delta)\mathrm{d}\Delta} \\
&= \lim_{T\to\infty} \frac{\int_O \frac{1}{\Delta}\tilde{\rho}_k^i(\Delta)\mathrm{d}\Delta}{\int_O \frac{1}{\Delta}\rho_k^i(\Delta)\mathrm{d}\Delta} \\
&\le \lim_{T\to\infty} \frac{U}{L} \cdot \frac{\int_{[T^{-p_0}, \frac{1}{1+\epsilon}]} \frac{1-(1-\epsilon\Delta)^{K-1}}{\Delta}\mathrm{d}\Delta + \int_{[\frac{1}{1+\epsilon}, 1]} \frac{1-\Delta^{K-1}}{\Delta}\mathrm{d}\Delta}{\int_O \frac{1-\Delta^{K-1}}{\Delta}\mathrm{d}\Delta} \\
&\le \lim_{T\to\infty} \frac{U}{L} \cdot \frac{\int_{[T^{-p_0}, \frac{1}{1+\epsilon}]} (K-1)\epsilon\mathrm{d}\Delta + \int_{[\frac{1}{1+\epsilon}, 1]} \frac{1}{\Delta}\mathrm{d}\Delta}{p_0 \log T - \frac{1}{K-1}(1 - T^{-p_0})} \\
&\le \lim_{T\to\infty} \frac{U}{L} \cdot \frac{(K-1)\epsilon + \log(1+\epsilon)}{p_0 \log T - \frac{1}{K-1}(1 - T^{-p_0})} = 0.
\end{aligned}
$$

Here the first equation is based on Eq.6, and the second last Inq. comes from Bernoulli inequality.

### B.3.3 PROOF OF LEMMA 7

To prove the lemma, We first reiterate an important lemma that lower bounds $\mathbb{E}[S_k^T(r, \mathcal{A})]$.

**Lemma 8** (Lemma 16.3 in Lattimore & Szepesvári (2020)). *Let $r, r' \in \Omega$ be 2 instances that differs only in one arm $k \in [K]$, where $\Delta_k > 0$ in $r$ and $k$ uniquely optimal in $r'$. Then for any algorithm $\mathcal{A}, \forall T,$*

$$\mathbb{E}[S_k^T(r, \mathcal{A})] \geq \frac{2}{(r_k - r'_k)^2}\Big[\log\Big(\frac{\min\{r'_k - r_k - \Delta_k, \Delta_k\}}{4}\Big) + \log T - \log(\text{Reg}_T(r, \mathcal{A}) + \text{Reg}_T(r', \mathcal{A}))\Big].$$

For fixed $\epsilon$ and $r$ we set $r'_k = r_k + (1 + \Delta_k)\epsilon$, so for $r \in \Lambda_k^\epsilon, r' \in \Omega$. On the other hand, when $T^{-p_0} > e(T), \forall r \in \Omega_T^k, s(\Delta_k, T) = \Delta_k + \frac{u(T)\log T}{\Delta_k}$. According to Lemma 8, for all $r \in \Lambda_k^\epsilon,$

$$\Delta_k \mathbb{E}[S_k^T(r, \tilde{\mathcal{A}})] \geq \frac{2}{\Delta_k(1 + \epsilon)^2}\Big[\log\Big(\frac{\epsilon\Delta_k}{4}\Big) + \log T - \log(\text{Reg}_T(r, \tilde{\mathcal{A}}) + \text{Reg}_T(r', \tilde{\mathcal{A}}))\Big].$$

Define $I_T^k \triangleq \int_{\Omega_T^k \cap \Lambda_k^\epsilon} \frac{p(r)}{\Delta_k}\log T \mathrm{d}r < \infty$, and we have $q_T^k(r) = \frac{p(r)/\Delta_k}{I_n^k}$ is thus a p.d.f. in $\Omega_T^k \cap \Lambda_k^\epsilon$. We already know that for any 1-subgaussian bandit instance, the regret of UCB algorithm is bounded by $8\sqrt{KT\log T} + 3\sum_{k=1}^K \Delta_k \leq 9\sqrt{KT\log T}$. The optimality of $\tilde{\mathcal{A}}(\mathbb{D}, T) = \arg\min_{\mathcal{A}} \text{Reg}_T(\mathbb{D}, \mathcal{A})$ implies that $\text{Reg}_T(\mathbb{D}, \tilde{\mathcal{A}}) \leq 9\sqrt{KT\log T}$. Due to the concavity of $\log(\cdot)$,

$$\int q(r)\log(\text{Reg}_T(r, \tilde{\mathcal{A}}) + \text{Reg}_T(r', \tilde{\mathcal{A}}))\mathrm{d}r$$

$$\leq \log \int q(r)(\text{Reg}_T(r, \tilde{\mathcal{A}}) + \text{Reg}_T(r', \tilde{\mathcal{A}}))\mathrm{d}r$$

$$= \log \int_{\Omega_T^k \cap \Lambda_k^\epsilon} \frac{p(r)}{\Delta_k}(\text{Reg}_T(r, \tilde{\mathcal{A}}) + \text{Reg}_T(r', \tilde{\mathcal{A}}))\mathrm{d}r - \log \int_{\Omega_T^k \cap \Lambda_k^\epsilon} \frac{p(r)}{\Delta_k}\mathrm{d}r$$

$$\leq p_0 \log T + \log \int_{\Omega_T^k \cap \Lambda_k^\epsilon} p(r)(\text{Reg}_T(r, \tilde{\mathcal{A}}) + \text{Reg}_T(r', \tilde{\mathcal{A}}))\mathrm{d}r - \log \int_{\Omega_T^k \cap \Lambda_k^\epsilon} p(r)\mathrm{d}r$$

$$\leq p_0 \log T + \log \Big( \text{Reg}_T(\mathbb{D}, \tilde{\mathcal{A}}) + \int \frac{q(r)}{q(r')}q(r')\text{Reg}_T(r', \tilde{\mathcal{A}}) \Big) - \log \int_{\Omega_T^k \cap \Lambda_k^\epsilon} p(r)\mathrm{d}r$$

$$\leq \log(\frac{9(L + U)\sqrt{K}}{L} \cdot T^{\frac{1}{2}+p_0}\log T) - \log \int_{\Omega_T^k \cap \Lambda_k^\epsilon} p(r)\mathrm{d}r.$$

Therefore, we have

$$\liminf_{T \to \infty} \frac{\int_{\Omega_T^k \cap \Lambda_k^\epsilon} p(r)\Delta_k \mathbb{E}[S_k^T(r, \tilde{\mathcal{A}})]\mathrm{d}r}{\int_{\Omega_T^k \cap \Lambda_k^\epsilon} p(r)s(\Delta_k, T)\mathrm{d}r} \tag{11}$$

$$\geq \frac{2}{(1 + \epsilon)^2} \liminf_{T \to \infty} \frac{\int_{\Omega_T^k \cap \Lambda_k^\epsilon} \frac{p(r)}{\Delta_k}\Big[\log\Big(\frac{\epsilon\Delta_k}{4}\Big) + \log T - \log(\text{Reg}_T(r, \tilde{\mathcal{A}}) + \text{Reg}_T(r', \tilde{\mathcal{A}}))\Big]\mathrm{d}r}{\int_{\Omega_T^k \cap \Lambda_k^\epsilon} \frac{p(r)}{\Delta_k}\big(\Delta_k^2 + u(T)\log T\big)\mathrm{d}r} \tag{12}$$

$$= \frac{1}{(1 + \epsilon)^2}\Big[1 + \liminf_{T \to \infty} \frac{1}{I_T^k}\int_{\Omega_T^k \cap \Lambda_k^\epsilon} \frac{p(r)}{\Delta_k}\Big(\log\Delta_k - \log(\text{Reg}_T(r, \tilde{\mathcal{A}}) + \text{Reg}_T(r', \tilde{\mathcal{A}}))\Big)\mathrm{d}r\Big] \tag{13}$$

$$\geq \frac{1}{(1 + \epsilon)^2}\Big[1 - p_0 - \limsup_{T \to \infty} \int_{\Omega_T^k \cap \Lambda_k^\epsilon} q_T^k(r)\log(\text{Reg}_T(r, \tilde{\mathcal{A}}) + \text{Reg}_T(r', \tilde{\mathcal{A}}))\mathrm{d}r\Big] \tag{14}$$

$$\geq \frac{1}{(1 + \epsilon)^2}\Big[1 - p_0 - \limsup_{T \to \infty} \frac{1}{\log T}\big(\log(\frac{9(L + U)\sqrt{K}}{L} \cdot T^{\frac{1}{2}+p_0}\log T) - \log \int_{\Omega_T^k \cap \Lambda_k^\epsilon} p(r)\mathrm{d}r\big)\Big] \tag{15}$$

$$= \frac{1}{(1 + \epsilon)^2}(\frac{1}{2} - 2p_0). \tag{16}$$

Here the last third line is because $r \geq T^{-p_0}, \forall r \in \Omega_T^k$.

### B.3.4 PROOF OF PART 1 IN THEOREM 1

**Lemma 9.** $\lim_{T\to\infty} \text{Reg}_T(\mathbb{D}, \tilde{\mathcal{A}}(\mathbb{D}, T)) = +\infty$.

*Proof.* In Lemma 7 we already show that

$$\liminf_{T\to\infty} \frac{\int_{\Omega_T^k \cap \Lambda_k^\epsilon} p(r)\Delta_k \mathbb{E}[S_k^T(r, \tilde{\mathcal{A}})]\mathrm{d}r}{\int_{\Omega_T^k \cap \Lambda_k^\epsilon} p(r)s(\Delta_k, T)\mathrm{d}r} \geq \frac{1}{(1+\epsilon)^2}(\frac{1}{2} - 2p_0) > 0.$$

To prove this lemma, if suffices to show that

$$\lim_{T\to\infty} \int_{\Omega_T^k \cap \Lambda_k^\epsilon} p(r)s(\Delta_k, T)\mathrm{d}r = +\infty.$$

Notice that $\Omega_T^k \cap \Lambda_k^\epsilon$ is non-decreasing in terms of $T$.

$$\begin{aligned}
\lim_{T\to\infty} \int_{\Omega_T^k \cap \Lambda_k^\epsilon} p(r)s(\Delta_k, T)\mathrm{d}r &\geq \lim_{T\to\infty} \int_{\Omega_T^k \cap \Lambda_k^\epsilon} L\frac{u(T)\log T}{\Delta_k}\mathrm{d}r \\
&\geq \lim_{T\to\infty} \int_{\Omega_T^k \cap \Lambda_k^\epsilon} Lu(T)\log T\mathrm{d}r \\
&\geq \lim_{T\to\infty} m(\omega_{T_0}^k \cap \Lambda_k^\epsilon)Lu(T)\log T = +\infty.
\end{aligned}$$

$\square$

## C OMITTED DETAILS FOR THEOREM 2

### C.1 SUBROUTINE FOR FINDING THE COVER SET

This subroutine is used to find a policy-value set $\hat{\Pi}$ such that $\hat{\Pi}$ covers $(1-3\delta)$-fraction of the MDPs in the sampled MDP set, i.e.

$$\frac{\sum_{i=1}^N \mathbb{I}[\exists(\pi, v) \in \hat{\Pi}, s.t.(\pi, v) \text{ covers } \mathcal{M}_i]}{N} \geq 1 - 3\delta.$$

The algorithm is a greedy algorithm consisting of at most $N$ steps. As the beginning of the algorithm, we calculate a matrix $\mathbf{A}$, where $\mathbf{A}_{i,j}$ indicates whether $(\pi_j, v_j)$ covers the MDP $\mathcal{M}_i$. In each step $t$, we find a policy-value pair $(\pi_{j_t}, v_{j_t})$ with the maximum cover number in the uncovered MDP set $\mathcal{T}_{t-1}$. We update the index set $\mathcal{U}_t$ and $\mathcal{T}_t$ according to the selected index $j_t$. We output the policy-value set $\hat{\Pi}$ once the cover size $\sum_{\tau=1}^t n_\tau \geq (1-3\delta)N$.

---

**Algorithm 4** Subroutine: Policy Cover Set

---

1: **Input**: $v_{i,j}$ for $i \in [N]$ and $j \in [N]$
2: Initialize: the policy index set $\mathcal{U}_0 = \emptyset$, the MDP index set $\mathcal{T}_0 = [N]$
3: Calculate the covering matrix $\mathbf{A} \in \mathbb{R}^{N \times N}$ where $\mathbf{A}_{i,j} = \text{Cnd}(v_{i,j}, v_{i,i}, v_{j,j})$
4: **for** $t = 1, \cdots, N$ **do**
5:     Calculate the policy index with maximum cover: $j_t = \arg\max_{[N]\setminus\mathcal{U}_{t-1}} \sum_{i\in\mathcal{T}_{t-1}} \mathbf{A}_{i,j}$
6:     Set $\mathcal{U}_t = \mathcal{U}_{t-1} \cup j_t, \mathcal{T}_t = \mathcal{T}_{t-1}\setminus\{i : \mathbf{A}_{i,j} = 1\}$, the cover size $n_t = \sum_{i\in\mathcal{T}_{t-1}} \mathbf{A}_{i,j_t}$
7:     **if** The cover size $\sum_{\tau=1}^t n_\tau \geq (1-3\delta)N$ **then**
8:         Denote $\mathcal{U}_t$ as $\mathcal{U}$, then break the loop
9: **Output**: the policy-value set $\hat{\Pi} = \{(\pi_j, v_{j,j}), \forall j \in \mathcal{U}\}$

---

## C.2 PROOF FOR THE PRE-TRAINING STAGE

During the proof, we use $\Omega^*$ to denote the MDP set satisfying the $(1 - \delta)$-cover condition with minimum cardinality, i.e. $\Omega^* = \arg\min_{\mathcal{P}(\tilde{\Omega}) \geq 1 - \delta} |\tilde{\Omega}|$. As defined in Algorithm 4, we use $\mathcal{U}$ to denote the index set of $\hat{\Omega}$, which has the same cardinality as $\hat{\Omega}$. We have the following lemma for the pre-training stage.

**Lemma 10** (Pre-training algorithm). *With probability at least $1 - \mathcal{O}\left(\delta \log \frac{\mathcal{C}(\mathbb{D})}{\delta}\right)$, the pre-training stage algorithm returns within $\log\left(\tilde{\mathcal{O}}(\mathcal{C}(\mathbb{D}))\right)$ phases with total MDP sample complexity bounded by $\tilde{\mathcal{O}}\left(\frac{\mathcal{C}(\mathbb{D})}{\delta^2}\right)$. The return set $\hat{\Pi}$ satisfies*

$$\Pr_{\mathcal{M} \sim \mathbb{D}} \left[ \exists (\pi, v) \in \hat{\Pi}, \left| V_{\mathcal{M},1}^{\pi}(s_1) - V_{\mathcal{M},1}^{*}(s_1) \right| < 2\epsilon \cap \left| V_{\mathcal{M},1}^{\pi}(s_1) - v \right| < 2\epsilon \right] \geq 1 - 6\delta, \quad (17)$$

*and the size is bounded by $|\hat{\Pi}| \leq 2\mathcal{C}(\mathbb{D}) \log \frac{1}{\delta}$.*

*Proof.* For each phase, we first define the high-probability event in Lemma 11. We prove the lemma in Appendix C.2.1.

**Lemma 11** (High Probability Events). *For all phases, with probability at least $1 - 4\delta$, the following events hold:*

1. $\left| \frac{1}{N} \sum_{i=1}^{N} \mathbb{I}\left[\mathcal{M}_i \in \Omega^*\right] - \mathcal{P}(\Omega^*) \right| \leq \delta.$

2. $\forall i \in [N], \pi_i$ is $\frac{\epsilon}{2}$-optimal for $\mathcal{M}_i$. $\forall i, j \in [N], \left| v_{i,j} - V_{\mathcal{M}_i,1}^{\pi_j}(s_1) \right| \leq \frac{\epsilon}{2}.$

3. *For all index set $\mathcal{U}' \subset [N]$, we have*

$$\Pr_{\mathcal{M} \sim \mathbb{D}} \left[ \exists j \in \mathcal{U}', \left| V_{\mathcal{M},1}^{\pi_j}(s_1) - V_{\mathcal{M},1}^{*}(s_1) \right| < 2\epsilon \cap \left| V_{\mathcal{M},1}^{\pi_j}(s_1) - v_{j,j} \right| < 2\epsilon \right] \quad (18)$$

$$\geq \frac{1}{N - |\mathcal{U}'|} \sum_{i \in [N] \setminus \mathcal{U}'} \max_{j \in \mathcal{U}'} \mathbf{A}_{i,j} - 2\delta - \sqrt{\frac{|\mathcal{U}'| \log \frac{2N}{\delta}}{N - |\mathcal{U}'|}}. \quad (19)$$

According to Lemma 11, the three events defined in Lemma 11 hold with probability $1 - 4\delta$. As stated in Lemma 12, condition on the first and second events, we have $|\mathcal{U}| \leq 2\mathcal{C}(\mathbb{D}) \log \frac{1}{\delta}$ for each phase. We prove Lemma 12 in Appendix C.2.2. Note that these lemmas also hold for the last phase in which we return the policy-value set $\hat{\Pi}$.

**Lemma 12.** *For all phases, if the first and second events in Lemma 11 hold, then the size of candidate set $\mathcal{U}$ satisfies*

$$|\mathcal{U}| \leq (\mathcal{C}(\mathbb{D}) + 1) \log(1/\delta). \quad (20)$$

Since the stopping condition is $\sqrt{\frac{|\hat{\Omega}| \log(2N/\delta)}{N - |\hat{\Omega}|}} = \sqrt{\frac{|\mathcal{U}| \log(2N/\delta)}{N - |\mathcal{U}|}} \leq \delta$ and we already know $|\mathcal{U}| \leq 2\mathcal{C}(D) \log \frac{1}{\delta}$, the stopping condition is satisfied for $N \geq \frac{4\mathcal{C}(\mathbb{D}) \log^2 (\mathcal{C}(\mathbb{D})/\delta)}{\delta^2}$. Based on the doubling trick, we know that the number of total phases is bounded by $\log\left(\tilde{\mathcal{O}}(\mathcal{C}(\mathbb{D}))\right)$ and the sample complexity is bounded by $2N = \tilde{\mathcal{O}}(\mathcal{C}(\mathbb{D})/\delta^2)$.

Therefore, by union bound across all phases, with probability at least $1 - \mathcal{O}(\log \frac{\mathcal{C}(\mathbb{D})}{\delta} \delta)$, Lemma 11 holds for all phases. Using the third event on the return set $\mathcal{U}$, we know that

$$\Pr_{\mathcal{M} \sim \mathbb{D}} \left[ \exists j \in \mathcal{U}, \left| V^{\pi_j}_{\mathcal{M},1}(s_1) - V^*_{\mathcal{M},1}(s_1) \right| < 2\epsilon \cap \left| V^{\pi_j}_{\mathcal{M},1}(s_1) - v_{j,j} \right| < 2\epsilon \right] \tag{21}$$

$$\geq \frac{1}{N - |\mathcal{U}|} \sum_{i \in [N] \setminus \mathcal{U}} \max_{j \in \mathcal{U}} \mathbf{A}_{i,j} - 2\delta - \sqrt{\frac{|\mathcal{U}| \log \frac{2N}{\delta}}{N - |\mathcal{U}|}} \tag{22}$$

$$\geq \frac{(1 - 3\delta)N - |\mathcal{U}|}{N - |\mathcal{U}|} - 2\delta - \delta \geq 1 - \frac{3\delta N}{N - |\mathcal{U}|} - 3\delta \geq 1 - 6\delta. \tag{23}$$

Therefore, the property on $\hat{\Pi}$ holds. $\qquad\square$

### C.2.1 Proof of Lemma 11

*Proof.* To prove this lemma, we sequentially bound the failure probability for each event.

**Empirical probability of $\Omega^*$.** Notice that since $\mathcal{M}_i \sim \mathbb{D}$, the expectation of r.v. $\mathbb{I}[\mathcal{M}_i \in \Omega^*]$ is exactly $\mathcal{P}(\Omega^*)$. According to the Chernoff Bound, we have

$$\Pr \left[ \left| \frac{1}{N} \sum_{i=1}^{N} \mathbb{I}[\mathcal{M}_i \in \Omega^*] - \mathcal{P}(\Omega^*) \right| \leq \delta \right] < \exp\{-2N\delta^2\} < \delta.$$

Therefore, the failure rate of the first event is bounded by $\delta$.

**Oracle error.** For each $i \in [N]$, the failure rate of $\mathbb{O}_l$ is at most $\frac{\delta}{N}$; For all $i, j \in [N]$, the failure rate of $\mathbb{O}_e$ is at most $\frac{\delta}{N^2}$. Therefore, with probability at least $1 - 2\delta$, we know that $\pi_i$ is indeed $\frac{\epsilon}{2}$-optimal for $\mathcal{M}_i$, and $\left| v_{ij} - V^{\pi_j}_{\mathcal{M}_i,1}(s_1) \right| < \frac{\epsilon}{2}$ for all $i, j \in [N]$. This implies that the failure rate of the second event is bounded by $2\delta$.

**Covering probability of $\mathcal{U}'$.** We first fix any index set $\mathcal{U}' \subset [N]$ and define $\mathcal{U}^c = [N] \setminus \mathcal{U}'$. For a policy-value pair $(\pi, v)$ and an MDP $\mathcal{M}$, we define the following random variable

$$\chi(\pi, v, \mathcal{M}) \triangleq \mathrm{Cnd}\Big( \mathbb{O}_e \left[ \mathcal{M}, \pi, \frac{\epsilon}{2}, \log(N^2/\delta)) \right], v,$$
$$\mathbb{O}_e \left[ \mathcal{M}, \mathbb{O}_l \left( \mathcal{M}, \frac{\epsilon}{2}, \log(N/\delta) \right), \frac{\epsilon}{2}, \log(N^2/\delta) \right] \Big).$$

Notice that $\mathbf{A}_{i,j}$ is exactly an instance of r.v. $\chi(\pi_j, v_j, \mathcal{M}_i)$. For a fixed index set $\mathcal{U}' \subset [N]$, each MDP $\mathcal{M}_i$ with index $i \in [N] \setminus \mathcal{U}'$ can be regarded as an i.i.d. sample from the distribution $\mathbb{D}$. According to Chernoff Bound, with probability at least $1 - \frac{\delta}{(2N)^{|\mathcal{U}'|}}$ we have

$$\frac{1}{|\mathcal{U}^c|} \sum_{i \in \mathcal{U}^c} \max_{j \in \mathcal{U}'} \mathbf{A}_{i,j} \leq \mathbb{E}_{\mathcal{M} \sim \mathbb{D}} \left[ \max_{j \in \mathcal{U}'} \{ \chi(\pi_j, v_j, \mathcal{M}) \} \right] + \sqrt{\frac{|\mathcal{U}'| \log \frac{2N}{\delta}}{|\mathcal{U}^c|}}. \tag{24}$$

On the other hand, we can use $\chi$ to control the probability that $\pi_j$ is near optimal for $\mathcal{M}_i$. Specifically,

$$\mathbb{E}_{\mathcal{M}\sim\mathbb{D}}\left[\max_{j\in\mathcal{U}'}\{\chi(\pi_j, v_j, \mathcal{M})\}\right] \tag{25}$$

$$= \Pr_{\mathcal{M}\sim\mathbb{D}}\left[\max_{j\in\mathcal{U}'}\{\chi(\pi_j, v_j, \mathcal{M})\} = 1\right] \tag{26}$$

$$= \sum_{\mathcal{M}\in\Omega}\mathcal{P}(\mathcal{M})\cdot\Pr\left[\max_{j\in\mathcal{U}'}\{\chi(\pi_j, v_j, \mathcal{M})\} = 1\big|\mathcal{M}\right] \tag{27}$$

$$= \sum_{\mathcal{M}\in\Omega}\mathcal{P}(\mathcal{M})\cdot\Pr\left[\exists j\in\mathcal{U}', \pi' = \mathbb{O}_l\left(\mathcal{M}, \frac{\epsilon}{2}, \log(N/\delta)\right),\right. \tag{28}$$

$$v = \mathbb{O}_e\left[\mathcal{M}, \pi, \frac{\epsilon}{2}, \log(N^2/\delta))\right], v' = \mathbb{O}_e\left[\mathcal{M}, \pi', \frac{\epsilon}{2}, \log(N^2/\delta)\right], \tag{29}$$

$$\left.\text{Cnd}(v, v', v_j) = 1\big|\mathcal{M}\right]. \tag{30}$$

Similar to the analysis in the second event of Lemma 11, with probability at least $1 - 2\delta$, for all $j \in \mathcal{U}'$, the return $\pi'$ is $\frac{\epsilon}{2}$-optimal for $\mathcal{M}$, and the estimated value $v$ and $v'$ in the RHS is $\frac{\epsilon}{2}$-close to their mean. Assume this event hold, we have

$$\mathbb{I}\left[\exists j\in\mathcal{U}', \text{Cnd}(v, v', v_j) = 1\right]$$
$$\leq \mathbb{I}\left[\exists j\in\mathcal{U}', \left|V_{\mathcal{M},1}^{\pi_j}(s_1) - V_{\mathcal{M},1}^*(s_1)\right| < 2\epsilon \cap \left|V_{\mathcal{M},1}^{\pi_j}(s_1) - v_{j,j}\right| < 2\epsilon\right].$$

Therefore, we can substitute the RHS in Eqn. 25 as (where $2\delta$ is the oracle failure probability)

$$\mathbb{E}_{\mathcal{M}\sim\mathbb{D}}\left[\max_{j\in\mathcal{U}'}\{\chi(\pi_j, v_j, \mathcal{M})\}\right] \tag{31}$$

$$\leq \sum_{\mathcal{M}\in\Omega}\mathcal{P}(\mathcal{M})\cdot\left((1 - 2\delta)\right. \tag{32}$$

$$\times \Pr\left[\exists j\in\mathcal{U}', \left|V_{\mathcal{M},1}^{\pi_j}(s_1) - V_{\mathcal{M},1}^*(s_1)\right| < 2\epsilon \cap \left|V_{\mathcal{M},1}^{\pi_j}(s_1) - v_{j,j}\right| < 2\epsilon|\mathcal{M}\right] + 2\delta\Big) \tag{33}$$

$$\leq 2\delta + \Pr_{\mathcal{M}\sim\mathbb{D}}\left[\exists j\in\mathcal{U}', \left|V_{\mathcal{M},1}^{\pi_j}(s_1) - V_{\mathcal{M},1}^*(s_1)\right| < 2\epsilon \cap \left|V_{\mathcal{M},1}^{\pi_j}(s_1) - v_{j,j}\right| < 2\epsilon\right]. \tag{34}$$

Combining Eqn. 24 and Eqn. 31, by the union bound, with probability at least

$$1 - \sum_{l=1}^N \frac{\delta}{(2N)^l}\left|\{\mathcal{U}' \subset [N], |\mathcal{U}'| = l\}\right| \geq 1 - \sum_{l=1}^N \frac{\delta}{(2N)^l}N^l \geq 1 - \delta,$$

for all $\mathcal{U}' \subset [N]$, we have

$$\Pr_{\mathcal{M}\sim\mathbb{D}}\left[\exists j\in\mathcal{U}', \left|V_{\mathcal{M},1}^{\pi_j}(s_1) - V_{\mathcal{M},1}^*(s_1)\right| < 2\epsilon \cap \left|V_{\mathcal{M},1}^{\pi_j}(s_1) - v_{j,j}\right| < 2\epsilon\right] \tag{35}$$

$$\geq \frac{1}{N - |\mathcal{U}'|}\sum_{i\in[N]\setminus\mathcal{U}'}\max_{j\in\mathcal{U}'}\mathbf{A}_{i,j} - 2\delta - \sqrt{\frac{|\mathcal{U}'|\log\frac{2N}{\delta}}{N - |\mathcal{U}'|}}. \tag{36}$$

$$\square$$

### C.2.2 PROOF OF LEMMA 12

*Proof.* For a certain fixed phase, we define $N_{\mathcal{M},t} = \sum_{i\in\mathcal{T}_t}\mathbb{I}\left[\mathcal{M}_i = \mathcal{M}\right]$ as the population of $\mathcal{M}$ in $\mathcal{T}_t$, and $\hat{C} \triangleq \sum_{i=1}^N \mathbb{I}\left[\mathcal{M}_i \in \Omega^*\right]$. We have $\hat{C} = \sum_{\mathcal{M}\in\Omega^*}N_{\mathcal{M},0}$. Thanks to the conditional events in Lemma 11, we have $\left|\frac{\hat{C}}{N} - \mathcal{P}(\Omega^*)\right| \leq \delta$ and $\frac{\hat{C}}{N} \geq 1 - 2\delta$.

Notice that $\mathcal{U}$ is generated by greedily selecting the best policy under current MDP remaining set $\mathcal{T}_{t-1}$, i.e. the policy that covers most MDP in $\mathcal{T}_{t-1}$. On the other hand, since the second event in

Lemma 11 holds, for $\mathcal{M}_i = \mathcal{M}_j$, we have $\text{Cnd}(v_{i,j}, v_{i,i}, v_{j,j})$ and $\text{Cnd}(v_{j,i}, v_{i,i}, v_{j,j})$ are true and thus $\mathbf{A}_{i,j} = \mathbf{A}_{j,i} = 1$. Therefore, for each step $t$ and each $\mathcal{M} \in \Omega^*$, if we have $\mathcal{M}_i = \mathcal{M}$ for some $i \in \mathcal{T}_{t-1}$, then in this round, we have

$$\sum_{i' \in \mathcal{T}_{t-1}} \mathbf{A}_{i',i} \geq N_{\mathcal{M}_i, t-1}.$$

Since we choose policy $\pi_{j_t}$ instead of $\pi_i$ in step $t$ according to the greedy strategy, we have

$$n_t = \sum_{i' \in \mathcal{T}_{t-1}} \mathbf{A}_{i',j_t} \geq \sum_{i' \in \mathcal{T}_{t-1}} \mathbf{A}_{i',i} \geq N_{\mathcal{M}, t-1} \tag{37}$$

for all $\mathcal{M} \in \Omega^*$. The right term is 0 if $\mathcal{M}$ no longer exists in the remaining set, and the inequality still holds. Summing over all $\mathcal{M} \in \Omega^*$, we obtain

$$n_t \geq \frac{1}{\mathcal{C}(\mathbb{D})} \sum_{\mathcal{M} \in \Omega^*} N_{\mathcal{M}, t-1}$$

$$= \frac{1}{\mathcal{C}(\mathbb{D})} \left( \sum_{\mathcal{M} \in \Omega^*} N_{\mathcal{M}, 0} - \sum_{\mathcal{M} \in \Omega^*} (N_{\mathcal{M}, 0} - N_{\mathcal{M}, t-1}) \right)$$

$$\geq \frac{1}{\mathcal{C}(\mathbb{D})} \left( \sum_{\mathcal{M} \in \Omega^*} N_{\mathcal{M}, 0} - \sum_{\mathcal{M} \in \Omega} (N_{\mathcal{M}, 0} - N_{\mathcal{M}, t-1}) \right)$$

$$= \frac{1}{\mathcal{C}(\mathbb{D})} \left( \sum_{\mathcal{M} \in \Omega^*} N_{\mathcal{M}, 0} - \sum_{\tau=1}^{t} n_\tau \right) = \frac{1}{\mathcal{C}(\mathbb{D})} \left( \hat{C} - \sum_{\tau=1}^{t} n_\tau \right),$$

where the last inequality is because $N_{\mathcal{M}, t}$ is monotonically decreasing in $t$ and the second last equation is because $\sum_{\tau=1}^{t} n_t$ represents the population of MDPs that are covered in the first $t$ rounds. This implies that (notice that $n_0 = 0$)

$$\hat{C} - \sum_{\tau=1}^{t} n_\tau \leq \frac{\mathcal{C}(\mathbb{D})}{\mathcal{C}(\mathbb{D}) + 1} \left( \hat{C} - \sum_{\tau=1}^{t-1} n_\tau \right) \leq \cdots \leq \left( \frac{\mathcal{C}(\mathbb{D})}{\mathcal{C}(\mathbb{D}) + 1} \right)^t \hat{C},$$

which gives

$$\sum_{\tau=1}^{t} n_\tau \geq \left( 1 - \left( \frac{\mathcal{C}(\mathbb{D})}{\mathcal{C}(\mathbb{D}) + 1} \right)^t \right) \hat{C}.$$

When $t \geq (\mathcal{C}(\mathbb{D}) + 1) \log \frac{1}{\delta}$, we have

$$|\mathcal{U}_t| = \sum_{\tau=1}^{t} n_\tau \geq \left( 1 - \exp\left\{ -\frac{t}{\mathcal{C}(\mathbb{D}) + 1} \right\} \right) \hat{C} \geq (1-\delta)N \times \frac{\hat{C}}{N} \geq (1-\delta)(1-2\delta) = 1 - 3\delta.$$

Therefore, upon breaking, the size of $\mathcal{U}$ satisfies $|\mathcal{U}| = |\mathcal{U}_t| \leq (\mathcal{C}(\mathbb{D}) + 1) \log \frac{1}{\delta}$. $\qquad\square$

## C.3 PROOF OF THEOREM 2

In Lemma 10, we prove that with probability at least $1 - \mathcal{O}\left( \delta \log \frac{\mathcal{C}(\mathbb{D})}{\delta} \right)$, the policy set $\hat{\Pi}$ returned in the pre-training stage covers the MDPs $\mathcal{M} \sim \mathbb{D}$ with probability at least $1 - 6\delta$, i.e.

$$\Pr_{\mathcal{M} \sim \mathbb{D}} \left[ \exists (\pi, v) \in \hat{\Pi}, \left| V_{\mathcal{M},1}^\pi(s_1) - V_{\mathcal{M},1}^*(s_1) \right| < 2\epsilon \cap \left| V_{\mathcal{M},1}^\pi(s_1) - v \right| < 2\epsilon \right] \geq 1 - 6\delta.$$

Note that this event happens with high probability. If this event does not happen (w.p. $\mathcal{O}\left( \delta \log \frac{\mathcal{C}(\mathbb{D})}{\delta} \right)$), the regret can still be upper bounded by $K$, which leads to an additional term of $K \cdot \mathcal{O}\left( \delta \log \frac{\mathcal{C}(\mathbb{D})}{\delta} \right) = \mathcal{O}\left( \sqrt{K} \log(K\mathcal{C}(\mathbb{D})) \right)$ in the final bound. This term is negligible compared with the dominant term in the regret. In the following analysis, we only discuss the case where the statement in Lemma 10 holds. We also assume that for the test MDP $M^*$,

$$\exists (\hat{\pi}^*, \hat{v}^*) \in \hat{\Pi}, \left| V_{\mathcal{M}^*,1}^{\hat{\pi}^*}(s_1) - V_{\mathcal{M}^*,1}^*(s_1) \right| < 2\epsilon \cap \left| V_{\mathcal{M}^*,1}^\pi(s_1) - \hat{v} \right| < 2\epsilon, \tag{38}$$

which will happen with probability $1 - 6\delta$ under the event defined in Lemma 10.

We use $L$ to denote the maximum epoch counter, which satisfies $L \leq |\hat{\Pi}|$.

**Lemma 13.** *(Optimism) With probability at least $1 - \delta/2$, we have $v_l \geq V^*_{\mathcal{M}^*,1}(s_1) - 2\epsilon, \forall l \in [L]$.*

*Proof.* To prove the lemma, we need to show that the optimal policy-value pair $(\hat{\pi}^*, \hat{v}^*)$ for $\mathcal{M}^*$ will never be eliminated from the set $\hat{\Pi}_l$ with high probability. Condition on the sampled $\mathcal{M}^*$, for a fixed episode $k \in [K]$, by Azuma's inequality, we have:

$$\Pr\left( \left| \frac{1}{k - k_0 + 1} \sum_{\tau=k_0}^{k} G_k - V^{\hat{\pi}^*}_{\mathcal{M}^*,1}(s_1) \right| \geq \sqrt{\frac{2\log(4K/\delta)}{k - k_0 + 1}} \right) \leq \delta/(2K). \quad (39)$$

By union bound over all $k \in [K]$, we know that

$$\Pr\left( \exists k \in [K], \left| \frac{1}{k - k_0 + 1} \sum_{\tau=k_0}^{k} G_k - V^{\hat{\pi}^*}_{\mathcal{M}^*,1}(s_1) \right| \geq \sqrt{\frac{2\log(4K/\delta)}{k - k_0 + 1}} \right) \leq \delta/2 \quad (40)$$

By Inq. 38, we know that $\left| V^{\hat{\pi}^*}_{\mathcal{M}^*,1}(s_1) - \hat{v}^* \right| < 4\epsilon$. Therefore,

$$\Pr\left( \exists k \in [K], \left| \frac{1}{k - k_0 + 1} \sum_{\tau=k_0}^{k} G_k - \hat{v}^* \right| \geq \sqrt{\frac{2\log(4K/\delta)}{k - k_0 + 1}} + 4\epsilon \right) \leq \delta/2 \quad (41)$$

By the elimination condition defined in line 6 of Algorithm 1 in the fine-tuning stage, $(\hat{\pi}^*, \hat{v}^*)$ will never be eliminated from the set $\hat{\Pi}_l$ with probability at least $1 - \delta/2$. By the definition that $v_l = \max_{(\pi,v) \in \hat{\Pi}_l} v$, we have $v_l \geq \hat{v}^* \geq V^*_{\mathcal{M}^*,1}(s_1) - 2\epsilon$.

$\square$

Now we are ready to prove Theorem 2.

*Proof.* We use $\tau_l$ to denote the starting episode of epoch $l$. Without loss of generality, we set $\tau_{L+1} = K + 1$.

By lemma 13, we know that $v_l \geq V^*_{\mathcal{M}^*,1}(s_1)$ with high probability. Under this event, we can decompose the value gap in the following way:

$$\sum_{k=1}^{k} \left( V^*_{\mathcal{M}^*,1}(s_1) - V^{\pi_k}_{\mathcal{M}^*,1}(s_1) \right) \leq \sum_{l=1}^{L} \sum_{\tau=\tau_l}^{\tau_{l+1}-1} \left( v_l - V^{\pi_l}_{\mathcal{M}^*,1}(s_1) \right) + 2\epsilon K$$

$$\leq \sum_{l=1}^{L} \sum_{\tau=\tau_l}^{\tau_{l+1}-1} (v_l - G_\tau) + \sum_{l=1}^{L} \sum_{\tau=\tau_l}^{\tau_{l+1}-1} \left( G_\tau - V^{\pi_l}_{\mathcal{M}^*,1}(s_1) \right) + 2\epsilon K$$

For the first term, by the elimination condition defined in line 6 of Algorithm 1 in the fine-tuning stage, we have

$$\sum_{\tau=\tau_l}^{\tau_{l+1}-1} (v_l - G_\tau) \leq \sqrt{2(\tau_{l+1} - \tau_l)\log(4K/\delta)} + 4(\tau_{l+1} - \tau_l)\epsilon + 1.$$

For the second term, by Azuma's inequality and union bound over all episodes, with probability at least $1 - \delta/2$,

$$\sum_{\tau=\tau_l}^{\tau_{l+1}-1} \left( G_\tau - V^{\pi_l}_{\mathcal{M}^*,1}(s_1) \right) \leq \sqrt{2(\tau_{l+1} - \tau_l)\log(4K/\delta)}$$

Therefore, by Cauchy-Schwarz inequality,

$$
\sum_{k=1}^{k} \left( V_{\mathcal{M}^*,1}^*(s_1) - V_{\mathcal{M}^*,1}^{\pi_k}(s_1) \right) \leq O \left( \sqrt{K|\hat{\Pi}| \log(4K/\delta)} + |\hat{\Pi}| \right)
$$

$$
\leq O \left( \sqrt{K\mathcal{C}(\mathbb{D}) \log(4K/\delta) \log(1/\delta)} + \mathcal{C}(\mathbb{D}) \right).
$$

The last inequality is due to $|\hat{\Pi}| \leq 2\mathcal{C}(\mathbb{D}) \log(1/\delta)$ by Lemma 10.

Finally, we take expectation over all possible $\mathcal{M}^*$, and we get

$$
\text{Reg}(K) \leq O \left( \sqrt{\mathcal{C}(\mathbb{D})K \log(4K/\delta) \log(1/\delta)} + \mathcal{C}(\mathbb{D}) \right). \tag{42}
$$

$\square$

### C.4 LOWER BOUND FOR THEOREM 2

In this subsection, we provide a lower bound to show that the regret upper bound in Theorem 2 is tight except for logarithmic factors. The lower bound is stated as follows.

**Theorem 14.** *Suppose $|\Omega| \geq 2$ and $K \geq 5$. For any pre-training and fine-tuning algorithm Alg, there exists a distribution $\mathbb{D}$ over the MDP class $\Omega$, such that the regret in the fine-tuning stage is at least*

$$
\text{Reg}_K(\mathbb{D}, Alg) \geq \Omega \left( \min \left( \sqrt{\mathcal{C}(\mathbb{D})K}, K \right) \right).
$$

This lower bound states that no matter how many samples are collected in the pre-training stage, the regret in the fine-tuning stage is at least $\Omega \left( \sqrt{\mathcal{C}(\mathbb{D})K} \right)$, which indicates that our upper bound is near-optimal except for logarithmic factors.

The proof is from an information theoretical perspective, which shares the similar idea with the lower bound proof for bandits (e.g. Theorem 15.2 in Lattimore & Szepesvári (2020) and Theorem 5.1 in Auer et al. (2002)). We first construct the hard instance and then prove the theorem. Since the bandit problem can be regarded as an MDP with horizon 1, our hard instance is constructed using a distribution over bandit instances. For a fixed algorithm Alg, we define $\Omega$ to be a set of $M$ multi-armed bandit instances. For bandit instance $\nu_i \in \Omega$, there are $M$ arms. The reward of arm $j$ is a Guassian distribution with unit variance and mean reward $\frac{1}{2} + \Delta\mathbb{I}\{i = j\}$. The parameter $\Delta \in [0, 1/2]$ will be defined later. We use $\nu_0$ to denote the bandit instance where the reward of each arm is a Guassian distribution with unit variance and mean reward $\frac{1}{2}$. We set $\mathbb{D}$ to be a uniform distribution over the MDP set $\Omega$.

We use $\mathbf{r}^k = \langle r_1, \cdots, r_k \rangle$ to denote the sequence of rewards received up through step $k \in [K]$. Note that any randomized algorithm Alg is equivalent to an a-prior random choice from the set of all deterministic algorithms. We can formally regard the algorithm Alg as a fixed function which maps the reward history $\mathbf{r}^{k-1}$ to the action $a_k$ in each step $k$. This technique is not crucial for the proof but simplifies the notations, which has also been applied in Auer et al. (2002).

With a slight abuse of notation, we use $\text{Reg}_K(\nu_i, Alg)$ to denote the regret of algorithm Alg in bandit instance $\nu_i$. Therefore, we have $\text{Reg}_K(\mathbb{D}, Alg) = \frac{1}{M} \sum_{i=1}^{M} \text{Reg}_K(\nu_i, Alg)$. We use $\mathbb{P}_{\nu_i}$ and $\mathbb{E}_{\nu_i}$ to denote the probability and the expectation under the condition that the bandit instance in the test stage is $\nu_i$, respectively. We use $D_{KL}(\mathbb{P}_1, \mathbb{P}_2)$ to denote the KL-divergence between the probability measure $\mathbb{P}_1$ and $\mathbb{P}_2$. We use $T_i(K)$ to denote the number of times arm $i$ is pulled in the $K$ steps.

We first provide the following lemma to upper bound the difference between expectations when measured using $\mathbb{E}_{\nu_i}$ and $\mathbb{E}_{\nu_0}$.

**Lemma 15.** *Let $f : \mathbb{R}^K \to [0, K]$ be any function defined on the reward sequence $\mathbf{r}$. Then for any action $i$,*

$$
\mathbb{E}_{\nu_i}[f(\mathbf{r})] \leq \mathbb{E}_{\nu_0}[f(\mathbf{r})] + \frac{K\Delta}{4} \sqrt{\mathbb{E}_{\nu_0}[T_i(K)]}.
$$

*Proof.* We upper bound the difference $\mathbb{E}_{\nu_i}[f(\mathbf{r})] - \mathbb{E}_{\nu_0}[f(\mathbf{r})]$ by calculating the expectation w.r.t. different probability measure.

$$\mathbb{E}_{\nu_i}[f(\mathbf{r})] - \mathbb{E}_{\nu_0}[f(\mathbf{r})] = \int_{\mathbf{r}} f(\mathbf{r}) d\mathbb{P}_{\nu_i}(\mathbf{r}) - \int_{\mathbf{r}} f(\mathbf{r}) d\mathbb{P}_{\nu_0}(\mathbf{r})$$

$$\leq \frac{K}{2} \left| \int_{\mathbf{r}} \left( d\mathbb{P}_{\nu_i}(\mathbf{r}) - d\mathbb{P}_{\nu_0}(\mathbf{r}) \right) \right|.$$

Here $\left| \int_{\mathbf{r}} \left( d\mathbb{P}_{\nu_i}(\mathbf{r}) - d\mathbb{P}_{\nu_0}(\mathbf{r}) \right) \right|$ is the TV-distance between the two probability measure $P_{\nu_i}(\mathbf{r})$ and $P_{\nu_0}(\mathbf{r})$. Note that $\nu_i$ and $\nu_0$ only differs in the expected reward of arm $i$. By Pinsker's inequality and Lemma 15.1 in Lattimore & Szepesvári (2020), we have

$$\left| \int_{\mathbf{r}} \left( d\mathbb{P}_{\nu_i}(\mathbf{r}) - d\mathbb{P}_{\nu_0}(\mathbf{r}) \right) \right| \leq \sqrt{\frac{1}{2} D_{KL}(\mathbb{P}_{\nu_i}, \mathbb{P}_{\nu_0})}$$

$$= \sqrt{\frac{1}{2} \sum_{k=1}^{K} \mathbb{P}_{\nu_0}(a_k = i) D_{KL}(\mathcal{N}(0,1) \| \mathcal{N}(\Delta, 1))}$$

$$= \sqrt{\mathbb{E}_{\nu_0}[T_i(K)] \frac{\Delta^2}{4}}.$$

The lemma can be proved by combining the above two inequalities. $\qquad\square$

Now we can prove Theorem 14.

*Proof.* By definition, we have

$$\text{Reg}_K(\mathbb{D}, \text{ Alg }) = \frac{1}{M} \sum_{i=1}^{M} \text{Reg}_K(\nu_i, \text{ Alg }) \tag{43}$$

$$= \Delta \frac{1}{M} \sum_{i=1}^{M} (K - \mathbb{E}_{\nu_i}[T_i(K)]) \tag{44}$$

$$= \Delta K - \frac{\Delta}{M} \sum_{i=1}^{M} \mathbb{E}_{\nu_i}[T_i(K)] \tag{45}$$

We apply Lemma 15 to $T_i(K)$, which is a function of the reward sequence $\mathbf{r}$ since the actions of the algorithm Alg are determined by the past rewards. We have

$$\mathbb{E}_{\nu_i}[T_i(K)] \leq \mathbb{E}_{\nu_0}[T_i(K)] + \frac{K\Delta}{4} \sqrt{\mathbb{E}_{\nu_0}[T_i(K)]}.$$

We sum the above inequality over all $\nu_i \in \Omega$, then we have

$$\sum_{i=1}^{M} \mathbb{E}_{\nu_i}[T_i(K)] \leq \sum_{i=1}^{M} \mathbb{E}_{\nu_0}[T_i(K)] + \sum_{i=1}^{M} \frac{K\Delta}{4} \sqrt{\mathbb{E}_{\nu_0}[T_i(K)]}$$

$$\leq K + \frac{K\Delta}{4} \sqrt{MK},$$

where the second inequality is due to the Cauchy-Schwarz inequality and the fact that $\sum_{\nu_i \in \Omega} \mathbb{E}_{\nu_0}[T_i(K)] = K$. Plgging this inequality back to Inq. 43, we have

$$\text{Reg}_K(\mathbb{D}, \text{ Alg }) \geq \Delta \left( K - \frac{K}{M} - \frac{K\Delta}{4} \sqrt{\frac{K}{M}} \right)$$

Since $K \geq 5$, we know that $\mathcal{C}(\mathbb{D})$ has the same order as $M$. If $K \leq M$, we know that $\sqrt{\mathcal{C}(\mathbb{D})K} = \Omega(K)$. We choose $\Delta = 1/2$ and know that $\text{Reg}_K(\mathbb{D}, \text{ Alg }) \geq \Omega(K)$. Therefore,

we have $\text{Reg}_K(\mathbb{D}, \text{Alg}) \geq \Omega\left(\min\left(\sqrt{\mathcal{C}(\mathbb{D})K}, K\right)\right)$. If $K \geq M$, we choose $\Delta = \frac{M}{K}$. Since $M \geq 2$, we can also prove that

$$\text{Reg}_K(\mathbb{D}, \text{Alg}) \geq \Omega\left(\min\left(\sqrt{\mathcal{C}(\mathbb{D})K}, K\right)\right).$$

$\square$

## D  OMITTED PROOF FOR THEOREM 3

We define $\pi^* = \arg\max_{\pi \in \Pi} \mathbb{E}_{\mathcal{M} \sim \mathbb{D}} V^\pi_{\mathcal{M},1}(s_1)$ and $\hat{\pi}^* = \arg\max_{\pi \in \Pi} \frac{1}{N} \sum_{i=1}^N V^\pi_{\mathcal{M}_i,1}(s_1)$. For the returned policy $\hat{\pi}$, we can decompose the value gap into the following terms:

$$\mathbb{E}_{\mathcal{M} \sim \mathbb{D}} \left[ V^{\pi^*}_{\mathcal{M},1}(s_1) - V^{\hat{\pi}}_{\mathcal{M},1}(s_1) \right] \leq \mathbb{E}_{\mathcal{M} \sim \mathbb{D}} V^{\pi^*}_{\mathcal{M},1}(s_1) - \frac{1}{N} \sum_{i=1}^N V^{\pi^*}_{\mathcal{M}_i,1}(s_1) \tag{46}$$

$$+ \frac{1}{N} \sum_{i=1}^N V^{\hat{\pi}}_{\mathcal{M}_i,1}(s_1) - \mathbb{E}_{\mathcal{M} \sim \mathbb{D}} V^{\hat{\pi}}_{\mathcal{M},1}(s_1) \tag{47}$$

$$+ \frac{1}{N} \sum_{i=1}^N V^{\hat{\pi}^*}_{\mathcal{M}_i,1}(s_1) - \frac{1}{N} \sum_{i=1}^N V^{\hat{\pi}}_{\mathcal{M}_i,1}(s_1) \tag{48}$$

$$+ \frac{1}{N} \sum_{i=1}^N V^{\pi^*}_{\mathcal{M}_i,1}(s_1) - \frac{1}{N} \sum_{i=1}^N V^{\hat{\pi}^*}_{\mathcal{M}_i,1}(s_1). \tag{49}$$

Note that the first and the second terms are generalization gap for a given policy, which can be upper bounded by Chernoff bound and union bound. The third term is the value gap during the training phase. The last term is less than $0$ by the optimality of $\hat{\pi}^*$.

**Upper bounds on the first and the second terms**  We can bound these terms following the generalization technique. Define the distance between polices $d(\pi^1, \pi^2) \triangleq \max_{s \in \mathcal{S}, h \in [H]} \|\pi_h^1(\cdot|s) - \pi_h^2(\cdot|s)\|_1$. We construct the $\epsilon_0$-covering set $\tilde{\Pi}$ w.r.t. $d$ such that

$$\forall \pi \in \Pi, \exists \tilde{\pi} \in \tilde{\Pi}, s.t. \quad d(\pi, \tilde{\pi}) \leq \epsilon_0. \tag{50}$$

By Inq. 50, we have

$$\forall \pi \in \Pi, \exists \tilde{\pi} \in \tilde{\Pi}, s.t. V^\pi_{\mathcal{M},1}(s_1) - V^{\tilde{\pi}}_{\mathcal{M},1}(s_1) \leq H\epsilon_0. \tag{51}$$

By definition of the covering number, $\left|\tilde{\Pi}\right| = \mathcal{N}(\Pi, \epsilon_0, d)$. By Chernoff bound and union bound over the policy set $\tilde{\Pi}$, we have with prob. $1 - \delta_1$, for any $\tilde{\pi} \in \tilde{\Pi}$,

$$\left| \frac{1}{N} \sum_{i=1}^N V^{\tilde{\pi}}_{\mathcal{M}_i,1}(s_1) - \mathbb{E}_{\mathcal{M} \sim \mathbb{D}} V^{\tilde{\pi}}_{\mathcal{M},1}(s_1) \right| \leq \sqrt{\frac{2 \log(2\mathcal{N}(\Pi, \epsilon_0, d)/\delta_1)}{N}}. \tag{52}$$

By Inq. 51 and Inq. 52, for $\forall \pi \in \Pi$,

$$\left| \frac{1}{N} \sum_{i=1}^N V^\pi_{\mathcal{M}_i,1}(s_1) - \mathbb{E}_{\mathcal{M} \sim \mathbb{D}} V^\pi_{\mathcal{M},1}(s_1) \right| \leq \left| \frac{1}{N} \sum_{i=1}^N V^{\tilde{\pi}}_{\mathcal{M}_i,1}(s_1) - \mathbb{E}_{\mathcal{M} \sim \mathbb{D}} V^{\tilde{\pi}}_{\mathcal{M},1}(s_1) \right| \tag{53}$$

$$+ \left| \frac{1}{N} \sum_{i=1}^N V^\pi_{\mathcal{M}_i,1}(s_1) - \frac{1}{N} \sum_{i=1}^N V^{\tilde{\pi}}_{\mathcal{M}_i,1}(s_1) \right| \tag{54}$$

$$+ \left| \mathbb{E}_{\mathcal{M} \sim \mathbb{D}} V^\pi_{\mathcal{M},1}(s_1) - \mathbb{E}_{\mathcal{M} \sim \mathbb{D}} V^{\tilde{\pi}}_{\mathcal{M},1}(s_1) \right| \tag{55}$$

$$\leq \sqrt{\frac{2 \log(2\mathcal{N}(\Pi, \epsilon_0, d)/\delta_1)}{N}} + 2H\epsilon_0. \tag{56}$$

We can set $\epsilon_0 = \frac{\epsilon}{12H}$ and $\delta_1 = 1/6$. Since $N = C_1 \log \left( \mathcal{N}\left( \Pi, \epsilon/(12H), d \right) \right)/\epsilon^2$ and $\pi^*, \hat{\pi} \in \Pi$, we know that with probability at least $5/6$,

$$\left| \frac{1}{N} \sum_{i=1}^{N} V_{\mathcal{M}_i,1}^{\pi^*}(s_1) - \mathbb{E}_{\mathcal{M} \sim \nu} V_{\mathcal{M},1}^{\pi^*}(s_1) \right| \leq \frac{\epsilon}{3}, \tag{57}$$

$$\left| \frac{1}{N} \sum_{i=1}^{N} V_{\mathcal{M}_i,1}^{\hat{\pi}}(s_1) - \mathbb{E}_{\mathcal{M} \sim \nu} V_{\mathcal{M},1}^{\hat{\pi}}(s_1) \right| \leq \frac{\epsilon}{3}. \tag{58}$$

**Upper bound on the third term** We have the following lemma, which is proved in the following subsections.

**Lemma 16.** *With probability at least $5/6$, Algorithm 2 can return a policy $\hat{\pi}$ satisfying*

$$\frac{1}{N} \sum_{i=1}^{N} V_{\mathcal{M}_i,1}^{\hat{\pi}^*}(s_1) - \frac{1}{N} \sum_{i=1}^{N} V_{\mathcal{M}_i,1}^{\hat{\pi}}(s_1) \leq \frac{\epsilon}{3},$$

*where $\hat{\pi}^*$ is the empirical maximizer, i.e. $\hat{\pi}^* = \arg\max_{\pi \in \Pi} \frac{1}{N} \sum_{i=1}^{N} V_{\mathcal{M}_i,1}^{\pi}(s_1)$.*

Pluging the results in Lemma 16, Inq. 57 and Inq. 58 back into Eq. 46, we know that with probability at least $2/3$,

$$\mathbb{E}_{\mathcal{M} \sim \mathbb{D}} \left[ V_{\mathcal{M},1}^{\pi^*}(s_1) - V_{\mathcal{M},1}^{\hat{\pi}}(s_1) \right] \leq \epsilon.$$

### D.1 PROOF OF LEMMA 16

We first state the following high-probability events.

**Lemma 17.** *With probability at least $1 - \frac{1}{2HK}$, the following inequality holds for any $i \in [N], k \in [K], h \in [H], s \in \mathcal{S}, a \in \mathcal{A}$,*

$$\left\| \hat{\mathbb{P}}_{\mathcal{M}_i,k,h}(\cdot|s,a) - \mathbb{P}_{\mathcal{M}_i,h}(\cdot|s,a) \right\|_1 \leq \sqrt{\frac{2S \log(8SANHK)}{\max\{1, N_{\mathcal{M}_i,k,h}(s,a)\}}} \tag{59}$$

$$\left| \hat{R}_{\mathcal{M}_i,k,h}(s,a) - r_{\mathcal{M}_i,h}(s,a) \right| \leq \sqrt{\frac{2 \log(8SANHK)}{\max\{1, N_{\mathcal{M}_i,k,h}(s,a)\}}} \tag{60}$$

*Proof.* According to Weissman et al. (2003), the $L_1$-deviation of the true distribution and the empirical distribution over $m$ distinct events from $n$ samples is bounded by

$$\mathbb{P}\left\{ \|\hat{p}(\cdot) - p(\cdot)\|_1 \geq \varepsilon \right\} \leq (2^m - 2) \exp\left( -\frac{n\varepsilon^2}{2} \right).$$

In our case where $m = S$, for any fixed $i, k, h, s, a$, we have

$$\left\| \hat{\mathbb{P}}_{\mathcal{M}_i,k,h}(\cdot|s,a) - \mathbb{P}_{\mathcal{M}_i,h}(\cdot|s,a) \right\|_1 \leq \sqrt{\frac{2S \log(1/\delta)}{\max\{1, N_{\mathcal{M}_i,k,h}(s,a)\}}} \tag{61}$$

with probability at least $1 - \delta$.

Taking union bound over all possible $i, k, h, s, a$, we know that Inq. 61 holds for any $i, k, h, s, a$ with probability at least $1 - NKHSA\delta$. We reach Inq. 59 by setting $\delta = \frac{1}{4NK^2H^2SA}$.

For the reward estimation, we know that the reward is 1-subgaussian by definition. By Hoeffding's inequality, we have

$$\left| \hat{R}_{\mathcal{M}_i,k,h}(s,a) - r_{\mathcal{M}_i}(s,a,h) \right| \leq \sqrt{\frac{2 \log(2/\delta)}{\max\{1, N_{\mathcal{M}_i,k,h}(s,a)\}}}$$

with probability at least $1 - \delta$ for any fixed $i, k, h, s, a$. Inq. 60 can be similarly proved by union bound over all possible $i, k, h, s, a$ and setting $\delta = \frac{1}{4NK^2H^2SA}$. $\qquad\square$

**Lemma 18.** *The following inequality holds with probability at least $1 - \frac{1}{2KH}$,*

$$\sum_{k=1}^{K}\sum_{i=1}^{N}\sum_{h=1}^{H}\left(\hat{V}_{\mathcal{M}_i,k,h}^{\pi_k}(s_{\mathcal{M}_i,k,h}) - V_{\mathcal{M}_i,h}^{\pi_k}(s_{\mathcal{M}_i,k,h})\right) - \left(\hat{Q}_{\mathcal{M}_i,k,h}^{\pi_k}(s_{\mathcal{M}_i,k,h}, a_{\mathcal{M}_i,k,h}) - Q_{\mathcal{M}_i,h}^{\pi_k}(s_{\mathcal{M}_i,k,h}, a_{\mathcal{M}_i,k,h})\right)$$

$$\leq \sqrt{2NHK\log(8KH)},$$

$$\sum_{k=1}^{K}\sum_{i=1}^{N}\sum_{h=1}^{H}\left(\mathbb{P}_{\mathcal{M}_i}\left(\hat{V}_{\mathcal{M}_i,k,h+1}^{\pi_k} - V_{\mathcal{M}_i,h}^{\pi_k}\right)(s_{\mathcal{M}_i,k,h}, a_{\mathcal{M}_i,k,h}) - \left(\hat{V}_{\mathcal{M}_i,k,h+1}^{\pi_k}(s_{k+1,h+1}) - V_{\mathcal{M}_i,h}^{\pi_k}(s_{k+1,h+1})\right)\right)$$

$$\leq \sqrt{2NHK\log(8KH)}$$

*Proof.* For the above two inequalities, the RHS can be regarded as the summation of Martingale differences. Therefore, the above inequalities hold by applying Azuma's inequality. □

We use $\Lambda_1$ to denote the events defined in the above lemmas. Now we prove the optimism of our algorithm under event $\Lambda_1$.

**Lemma 19.** *Under event $\Lambda_1$, we have $\hat{V}_{\mathcal{M}_i,k,1}^{\pi}(s_1) \geq V_{\mathcal{M}_i,1}^{\pi}(s_1)$ for any $i \in [N], k \in [K], \pi \in \Pi$.*

*Proof.* We prove the lemma by induction. Suppose $\hat{Q}_{\mathcal{M},k,h+1}^{\pi}(s,a) \geq Q_{\mathcal{M},h+1}^{\pi}(s,a)$. For $h \in [H]$, if $\hat{Q}_{\mathcal{M},k,h}^{\pi}(s,a) = 1$, then we trivially have $\hat{Q}_{\mathcal{M},k,h}^{\pi}(s,a) \geq Q_{\mathcal{M},h}^{\pi}(s,a)$. If $\hat{Q}_{\mathcal{M},k,h}^{\pi}(s,a) < 1$, then

$$\hat{Q}_{\mathcal{M},k,h}^{\pi}(s,a) - Q_{\mathcal{M},h}^{\pi}(s,a)$$

$$\geq \hat{R}_{\mathcal{M}_i,k,h}(s,a) + b_{\mathcal{M}_i,k,h}(s,a) + \hat{\mathbb{P}}_{\mathcal{M}_i,k,h}\hat{V}_{\mathcal{M}_i,k,h+1}^{\pi}(s,a) - r_{\mathcal{M}_i,h}(s,a) - \mathbb{P}_{\mathcal{M}_i,h}V_{\mathcal{M}_i,h+1}^{\pi}(s,a)$$

$$\geq b_{\mathcal{M}_i,k,h}(s,a) - \left|\hat{R}_{\mathcal{M}_i,k,h}(s,a) - r_{\mathcal{M}_i,h}(s,a)\right| - \left\|\hat{\mathbb{P}}_{\mathcal{M}_i,k,h}(\cdot|s,a) - \mathbb{P}_{\mathcal{M}_i,h}(\cdot|s,a)\right\|_1$$

$$\quad + \mathbb{P}_{\mathcal{M}_i,h}\left(\hat{V}_{\mathcal{M}_i,k,h+1}^{\pi} - V_{\mathcal{M}_i,h+1}^{\pi}\right)(s,a)$$

$$\geq b_{\mathcal{M}_i,k,h}(s,a) - \sqrt{\frac{2\log(8SANHK)}{\max\{1, N_{\mathcal{M}_i,k,h}(s,a)\}}} - \sqrt{\frac{2S\log(8SANHK)}{\max\{1, N_{\mathcal{M}_i,k,h}(s,a)\}}}$$

$$\quad + \mathbb{P}_{\mathcal{M}_i,h}\left(\hat{V}_{\mathcal{M}_i,k,h+1}^{\pi} - V_{\mathcal{M}_i,h+1}^{\pi}\right)(s,a)$$

$$\geq \mathbb{P}_{\mathcal{M}_i,h}\left(\hat{V}_{\mathcal{M}_i,k,h+1}^{\pi} - V_{\mathcal{M}_i,h+1}^{\pi}\right)(s,a)$$

$$\geq 0,$$

where the second inequality is due to Lemma 17 and the third inequality is derived from the definition of $b_{\mathcal{M}_i,k,h}(s,a)$. The last inequality is from the induction condition that $\hat{Q}_{\mathcal{M},k,h+1}^{\pi}(s,a) \geq Q_{\mathcal{M},h+1}^{\pi}(s,a)$. Therefore, for step $h$, we also have $\hat{Q}_{\mathcal{M},k,h}^{\pi}(s,a) \geq Q_{\mathcal{M},h}^{\pi}(s,a)$.

From the induction, we know that $\hat{Q}_{\mathcal{M},k,1}^{\pi}(s,a) \geq Q_{\mathcal{M},1}^{\pi}(s,a)$. By definition, we have $\hat{V}_{\mathcal{M}_i,k,1}^{\pi}(s_1) \geq V_{\mathcal{M}_i,1}^{\pi}(s_1)$. □

Now we prove Lemma 16.

*Proof.* (Proof of Lemma 16) By Lemma 19 and the optimality of $\hat{\pi}_k$,

$$\sum_{k=1}^{K}\sum_{i=1}^{N}\left(V_{\mathcal{M}_i,1}^{\hat{\pi}^*}(s_1) - V_{\mathcal{M}_i,1}^{\pi_k}(s_1)\right) \leq \sum_{k=1}^{K}\sum_{i=1}^{N}\left(\hat{V}_{\mathcal{M}_i,k,1}^{\pi_k}(s_1) - V_{\mathcal{M}_i,1}^{\pi_k}(s_1)\right). \tag{62}$$

We decompose the value difference using Bellman equation. For each episode $k \in [K]$:

$$
\begin{aligned}
&\hat{V}^{\pi_k}_{\mathcal{M}_i,k,h}(s_{\mathcal{M}_i,k,h}) - V^{\pi_k}_{\mathcal{M}_i,h}(s_{\mathcal{M}_i,k,h}) \\
={} &\left(\hat{V}^{\pi_k}_{\mathcal{M}_i,k,h}(s_{\mathcal{M}_i,k,h}) - V^{\pi_k}_{\mathcal{M}_i,h}(s_{\mathcal{M}_i,k,h})\right) - \left(\hat{Q}^{\pi_k}_{\mathcal{M}_i,k,h}(s_{\mathcal{M}_i,k,h}, a_{\mathcal{M}_i,k,h}) - Q^{\pi_k}_{\mathcal{M}_i,h}(s_{\mathcal{M}_i,k,h}, a_{\mathcal{M}_i,k,h})\right) \\
&+ \hat{Q}^{\pi_k}_{\mathcal{M}_i,k,h}(s_{\mathcal{M}_i,k,h}, a_{\mathcal{M}_i,k,h}) - Q^{\pi_k}_{\mathcal{M}_i,h}(s_{\mathcal{M}_i,k,h}, a_{\mathcal{M}_i,k,h}) \\
\leq{} &\left(\hat{V}^{\pi_k}_{\mathcal{M}_i,k,h}(s_{\mathcal{M}_i,k,h}) - \hat{Q}^{\pi_k}_{\mathcal{M}_i,k,h}(s_{\mathcal{M}_i,k,h}, a_{\mathcal{M}_i,k,h})\right) - \left(V^{\pi_k}_{\mathcal{M}_i,h}(s_{\mathcal{M}_i,k,h}) - Q^{\pi_k}_{\mathcal{M}_i,h}(s_{\mathcal{M}_i,k,h}, a_{\mathcal{M}_i,k,h})\right) \\
&+ \hat{R}_{\mathcal{M}_i,k,h}(s_{\mathcal{M}_i,k,h}, a_{\mathcal{M}_i,k,h}) - r_{\mathcal{M}_i,h}(s_{\mathcal{M}_i,k,h}, a_{\mathcal{M}_i,k,h}) + \left(\hat{\mathbb{P}}_{\mathcal{M}_i,k,h} - \mathbb{P}_{\mathcal{M}_i,h}\right)\hat{V}^{\pi_k}_{\mathcal{M}_i,k,h+1}(s_{\mathcal{M}_i,k,h}, a_{\mathcal{M}_i,k,h}) \\
&+ \mathbb{P}_{\mathcal{M}_i,h}\left(\hat{V}^{\pi_k}_{\mathcal{M}_i,k,h+1} - V^{\pi_k}_{\mathcal{M}_i,h}\right)(s_{\mathcal{M}_i,k,h}, a_{\mathcal{M}_i,k,h}) - \left(\hat{V}^{\pi_k}_{\mathcal{M}_i,k,h+1}(s_{\mathcal{M}_i,k+1,h+1}) - V^{\pi_k}_{\mathcal{M}_i,h}(s_{\mathcal{M}_i,k+1,h+1})\right) \\
&+ b_{\mathcal{M}_i,k,h}(s_{\mathcal{M}_i,k,h}, a_{\mathcal{M}_i,k,h}) \\
&+ \hat{V}^{\pi_k}_{\mathcal{M}_i,k,h+1}(s_{\mathcal{M}_i,k+1,h+1}) - V^{\pi_k}_{\mathcal{M}_i,h}(s_{\mathcal{M}_i,k+1,h+1}).
\end{aligned}
$$

Therefore,

$$
\begin{aligned}
&\sum_{k=1}^{K}\sum_{i=1}^{N}\left(V^{\hat{\pi}^*}_{\mathcal{M}_i,1}(s_1) - V^{\pi_k}_{\mathcal{M}_i,1}(s_1)\right) \\
\leq{} &\sum_{k=1}^{K}\sum_{i=1}^{N}\sum_{h=1}^{H}\left(\hat{V}^{\pi_k}_{\mathcal{M}_i,k,h}(s_{\mathcal{M}_i,k,h}) - V^{\pi_k}_{\mathcal{M}_i,h}(s_{\mathcal{M}_i,k,h})\right) - \left(\hat{Q}^{\pi_k}_{\mathcal{M}_i,k,h}(s_{\mathcal{M}_i,k,h}, a_{\mathcal{M}_i,k,h}) - Q^{\pi_k}_{\mathcal{M}_i,h}(s_{\mathcal{M}_i,k,h}, a_{\mathcal{M}_i,k,h})\right) \\
&+ \sum_{k=1}^{K}\sum_{i=1}^{N}\sum_{h=1}^{H}\left(\hat{R}_{\mathcal{M}_i,k,h}(s_{\mathcal{M}_i,k,h}, a_{\mathcal{M}_i,k,h}) - r_{\mathcal{M}_i,h}(s_{\mathcal{M}_i,k,h}, a_{\mathcal{M}_i,k,h})\right) \\
&+ \sum_{k=1}^{K}\sum_{i=1}^{N}\sum_{h=1}^{H}\left(\hat{\mathbb{P}}_{\mathcal{M}_i,k,h} - \mathbb{P}_{\mathcal{M}_i,h}\right)\hat{V}^{\pi_k}_{\mathcal{M}_i,k,h+1}(s_{\mathcal{M}_i,k,h}, a_{\mathcal{M}_i,k,h}) \\
&+ \sum_{k=1}^{K}\sum_{i=1}^{N}\sum_{h=1}^{H} b_{\mathcal{M}_i,k,h}(s_{\mathcal{M}_i,k,h}, a_{\mathcal{M}_i,k,h}) \\
&+ \sum_{k=1}^{K}\sum_{i=1}^{N}\sum_{h=1}^{H}\left(\mathbb{P}_{\mathcal{M}_i,h}\left(\hat{V}^{\pi_k}_{\mathcal{M}_i,k,h+1} - V^{\pi_k}_{\mathcal{M}_i,h}\right)(s_{\mathcal{M}_i,k,h}, a_{\mathcal{M}_i,k,h}) - \left(\hat{V}^{\pi_k}_{\mathcal{M}_i,k,h+1}(s_{k+1,h+1}) - V^{\pi_k}_{\mathcal{M}_i,h}(s_{k+1,h+1})\right)\right) \\
\leq{} &\sum_{k=1}^{K}\sum_{i=1}^{N}\sum_{h=1}^{H}\left(\hat{R}_{\mathcal{M}_i,k,h}(s_{\mathcal{M}_i,k,h}, a_{\mathcal{M}_i,k,h}) - r_{\mathcal{M}_i,h}(s_{\mathcal{M}_i,k,h}, a_{\mathcal{M}_i,k,h}) + \left(\hat{\mathbb{P}}_{\mathcal{M}_i,k,h} - \mathbb{P}_{\mathcal{M}_i,h}\right)\hat{V}^{\pi_k}_{\mathcal{M}_i,k,h+1}(s_{\mathcal{M}_i,k,h}, a_{\mathcal{M}_i,k,h})\right) \\
&+ \sum_{k=1}^{K}\sum_{i=1}^{N}\sum_{h=1}^{H} b_{\mathcal{M}_i,k,h}(s_{\mathcal{M}_i,k,h}, a_{\mathcal{M}_i,k,h}) + O(\sqrt{NHK\log(KH)}).
\end{aligned}
$$

The last inequality is due to Lemma 19 under event $\Lambda_1$. By Lemma 17, we have

$$
\begin{aligned}
&\sum_{k=1}^{K}\sum_{i=1}^{N}\sum_{h=1}^{H}\left(\hat{R}_{\mathcal{M}_i,k,h}(s_{\mathcal{M}_i,k,h}, a_{\mathcal{M}_i,k,h}) - r_{\mathcal{M}_i,h}(s_{\mathcal{M}_i,k,h}, a_{\mathcal{M}_i,k,h}) + \left(\hat{\mathbb{P}}_{\mathcal{M}_i,k,h} - \mathbb{P}_{\mathcal{M}_i,h}\right)\hat{V}^{\pi_k}_{\mathcal{M}_i,k,h+1}(s_{\mathcal{M}_i,k,h}, a_{\mathcal{M}_i,k,h})\right) \\
\leq{} &\sum_{k=1}^{K}\sum_{i=1}^{N}\sum_{h=1}^{H}\left(\hat{R}_{\mathcal{M}_i,k,h}(s_{\mathcal{M}_i,k,h}, a_{\mathcal{M}_i,k,h}) - r_{\mathcal{M}_i,h}(s_{\mathcal{M}_i,k,h}, a_{\mathcal{M}_i,k,h})\right) \\
&+ \sum_{k=1}^{K}\sum_{i=1}^{N}\sum_{h=1}^{H}\left\|\hat{\mathbb{P}}_{\mathcal{M}_i,k,h}(\cdot|s_{\mathcal{M}_i,k,h}, a_{\mathcal{M}_i,k,h}) - \mathbb{P}_{\mathcal{M}_i,h}(\cdot|s_{\mathcal{M}_i,k,h}, a_{\mathcal{M}_i,k,h})\right\|_1 \\
\leq{} &b_{\mathcal{M}_i,k,h}(s_{\mathcal{M}_i,k,h}, a_{\mathcal{M}_i,k,h})
\end{aligned}
$$

We can upper bound $\sum_{k=1}^{K} \sum_{i=1}^{N} \left( V_{\mathcal{M}_i,1}^{\hat{\pi}^*}(s_1) - V_{\mathcal{M}_i,1}^{\pi_k}(s_1) \right)$ by the summation of $b_{\mathcal{M}_i,k,h}(s_{\mathcal{M}_i,k,h}, a_{\mathcal{M}_i,k,h})$. By definition, we have

$$\sum_{k=1}^{K} \sum_{i=1}^{N} \left( V_{\mathcal{M}_i,1}^{\hat{\pi}^*}(s_1) - V_{\mathcal{M}_i,1}^{\pi_k}(s_1) \right) \leq \sum_{k=1}^{K} \sum_{i=1}^{N} \sum_{h=1}^{H} \sqrt{\frac{2S \log(8SANHK)}{\max\{1, N_{\mathcal{M}_i,k,h}(s_{\mathcal{M}_i,k,h}, a_{\mathcal{M}_i,k,h})\}}} + O(\sqrt{NHK \log(KH)}).$$

By Cauchy-Schwarz inequality, we have

$$\sum_{k=1}^{K} \sum_{i=1}^{N} \left( V_{\mathcal{M}_i,1}^{\hat{\pi}^*}(s_1) - V_{\mathcal{M}_i,1}^{\pi_k}(s_1) \right) \leq O\left( N\sqrt{H^2 S^2 A K \log(SAHNK)} + NHS^2 A \right).$$

Since $\hat{\pi}$ is uniformly selected from the policy set $\{\pi_k\}_{k=1}^K$. By Markov's inequality, the following inequality holds with probability at least $5/6$,

$$\sum_{i=1}^{N} \left( V_{\mathcal{M}_i,1}^{\hat{\pi}^*}(s_1) - V_{\mathcal{M}_i,1}^{\hat{\pi}}(s_1) \right) \leq \frac{1}{6K} \sum_{k=1}^{K} \sum_{i=1}^{N} \left( V_{\mathcal{M}_i,1}^{\hat{\pi}^*}(s_1) - V_{\mathcal{M}_i,1}^{\pi_k}(s_1) \right).$$

With our choice of $K = C_2 S^2 A H^2 \log(SAH/\epsilon)/\epsilon^2$, we have

$$\frac{1}{N} \sum_{i=1}^{N} \left( V_{\mathcal{M}_i,1}^{\hat{\pi}^*}(s_1) - V_{\mathcal{M}_i,1}^{\hat{\pi}}(s_1) \right) \leq \epsilon/3.$$

$\square$

### D.2 HIGH PROBABILITY BOUND

To obtain a high probability bound with probability at least $1 - \delta$. Our idea is to first execute Algorithm 2 independently for $O(\log(1/\delta))$ times and obtains a policy set with cardinality $O(\log(1/\delta))$. We evaluate the policies in the policy set on the sampled $M$ MDPs, and then return the policy with the maximum empirical value. The algorithm is described in Algorithm 5

---

**Algorithm 5** OMERM with High Probability

**Input**: target accuracy $\epsilon > 0$, high probability parameter $\delta$
$N = C_1 \log \left( \mathcal{N} \left( \Pi, \epsilon/(12H), d \right) / \delta \right) / \epsilon^2$ for a constant $C_1 > 0$
$N_1 = \log(2/\delta) / \log(1/6)$, $N_2 = C_2 \log(NN_1/\delta)/\epsilon^2$ for a constant $C_2 > 0$
Sample $N$ tasks from the distribution $\mathbb{D}$, denoted as $\{\mathcal{M}_1, \mathcal{M}_2, \cdots, \mathcal{M}_N\}$
5: **for** $\xi = 1, 2, \cdots, N_1$ **do**
    Execute Algorithm 2 with target accuracy $\epsilon/2$ and task set $\{\mathcal{M}_1, \mathcal{M}_2, \cdots, \mathcal{M}_N\}$, and obtain a policy $\pi_\xi$
    **for** task index $i = 1, 2, \cdots, N$ **do**
        Execute $\pi_\xi$ on task $\mathcal{M}_i$ for $N_2$ times, denoted the average total rewards as $V_{\xi,i}$
    Calculate the average value $V_\xi = \frac{1}{N} \sum_{i=1}^{N} V_{\xi,i}$
10: **Output**: the policy $\pi_{\xi^*}$ with $\xi^* = \arg\max_{\xi \in [N_1]} V_\xi$

---

We have the following theorem for Algorithm 5.

**Theorem 20.** *With probability at least $1 - \delta$, Algorithm 5 can output a policy $\hat{\pi}$ satisfying* $\mathbb{E}_{\mathcal{M}^* \sim \mathbb{D}}[V_{\mathcal{M}^*}^{\pi^*(\mathbb{D})} - V_{\mathcal{M}^*}^{\hat{\pi}}] \leq \epsilon$ *with* $\mathcal{O}\left( \frac{\log\left(\mathcal{N}_{\epsilon/(12H)}^{\Pi}/\delta\right)}{\epsilon^2} \right)$ *MDP instance samples during training. The number of episodes collected for each task is bounded by* $\mathcal{O}\left( \frac{H^2 S^2 A \log(SAH) \log(1/\delta)}{\epsilon^2} \right)$.

The proof follows the proof idea of Theorem 3, with only the difference in the bound on the empirical risk. We first prove the following lemma.

**Lemma 21.** *With probability at least $1 - \delta_2$, Algorithm 5 can return a policy $\pi_{\xi^*}$ satisfying*

$$\frac{1}{N} \sum_{i=1}^{N} V_{\mathcal{M}_i,1}^{\hat{\pi}^*}(s_1) - \frac{1}{N} \sum_{i=1}^{N} V_{\mathcal{M}_i,1}^{\pi_{\xi^*}}(s_1) \leq \frac{\epsilon}{3},$$

*where $\hat{\pi}^*$ is the empirical maximizer, i.e. $\hat{\pi}^* = \arg\max_{\pi \in \Pi} \frac{1}{N} \sum_{i=1}^{N} V_{\mathcal{M}_i,1}^{\pi}(s_1)$.*

*Proof.* By Lemma 16, for each $\xi \in [N_1]$, the following inequality holds with probability at least $5/6$,

$$\frac{1}{N} \sum_{i=1}^{N} V_{\mathcal{M}_i,1}^{\hat{\pi}^*}(s_1) - \frac{1}{N} \sum_{i=1}^{N} V_{\mathcal{M}_i,1}^{\pi_\xi}(s_1) \leq \frac{\epsilon}{9}. \tag{63}$$

In Algorithm 5, we evaluate the policies in the MDPs by executing the policies for $N_2$ times. By Hoeffding's inequality and the union bound over all $\xi \in [N_1]$ and $i \in [N]$, with probability at least $1 - \delta_2/2$, we have

$$\left| V_\xi - \frac{1}{N} \sum_{i=1}^{N} V_{\mathcal{M}_i,1}^{\pi_\xi}(s_1) \right| \leq \frac{\epsilon}{9}, \forall \xi \in [N_1], i \in [N]$$

We denote the above event as $\Lambda_2$. For each $\xi \in [N_1]$, we define $\eta_i$ as the event that $\frac{1}{N} \sum_{i=1}^{N} V_{\mathcal{M}_i,1}^{\hat{\pi}^*}(s_1) - V_\xi \leq \frac{2\epsilon}{9}$. By Inq. 63, we have $\Pr\{\eta_i\} \geq 5/6$ under event $\Lambda_2$. Note that the event $\{\eta_i\}_{i=1}^{N_1}$ are independent with each other. Therefore, with probability at least $1 - (1/6)^{N_1} = 1 - \delta_2/2$, there exists $\xi_0$, such that the event $\eta_{\xi_0}$ happens. That is, there exists a policy $\pi_{\xi_0}$ such that

$$\frac{1}{N} \sum_{i=1}^{N} V_{\mathcal{M}_i,1}^{\hat{\pi}^*}(s_1) - V_{\xi_0} \leq \frac{2\epsilon}{9}.$$

By definition, $\xi^* = \arg\max_{\xi \in [N_1]} V_\xi$. Therefore, we have $\frac{1}{N} \sum_{i=1}^{N} V_{\mathcal{M}_i,1}^{\hat{\pi}^*}(s_1) - V_{\xi^*} \leq \frac{2\epsilon}{9}$. Under the event of $\Lambda_2$, we have the following inequality holds with probability at least $1 - \delta_2/2$,

$$\frac{1}{N} \sum_{i=1}^{N} V_{\mathcal{M}_i,1}^{\hat{\pi}^*}(s_1) - \frac{1}{N} \sum_{i=1}^{N} V_{\mathcal{M}_i,1}^{\pi_{\xi^*}}(s_1) \leq \frac{\epsilon}{3}.$$

$\square$

*Proof.* (Proof of Theorem 20) The proof follows the proof idea of Theorem 3. We bound the empirical risk by Lemma 21. With our choice of $\delta_2 = \delta/3$, we have with probability at least $1 - \delta/2$,

$$\frac{1}{N} \sum_{i=1}^{N} V_{\mathcal{M}_i,1}^{\hat{\pi}^*}(s_1) - \frac{1}{N} \sum_{i=1}^{N} V_{\mathcal{M}_i,1}^{\pi_{\xi^*}}(s_1) \leq \frac{\epsilon}{3}. \tag{64}$$

Following the proof of Theorem 3, we can similarly show that the following inequalities hold with probability $1 - \delta/2$,

$$\left| \frac{1}{N} \sum_{i=1}^{N} V_{\mathcal{M}_i,1}^{\pi^*}(s_1) - \mathbb{E}_{\mathcal{M} \sim \nu} V_{\mathcal{M},1}^{\pi^*}(s_1) \right| \leq \frac{\epsilon}{3}, \tag{65}$$

$$\left| \frac{1}{N} \sum_{i=1}^{N} V_{\mathcal{M}_i,1}^{\hat{\pi}}(s_1) - \mathbb{E}_{\mathcal{M} \sim \nu} V_{\mathcal{M},1}^{\hat{\pi}}(s_1) \right| \leq \frac{\epsilon}{3}. \tag{66}$$

Combining the results in Inq. 65, Inq.66 and Inq. 64, we know that with probability at least $1 - \delta$,

$$\mathbb{E}_{\mathcal{M} \sim \mathbb{D}} \left[ V_{\mathcal{M},1}^{\pi^*}(s_1) - V_{\mathcal{M},1}^{\hat{\pi}}(s_1) \right] \leq \epsilon.$$

$\square$

