# OpenReview forum: "On the Power of Pre-training for Generalization in RL: Provable Benefits and Hardness"
_ICLR.cc/2023/Conference — Submitted to ICLR 2023_

### Official Review · Reviewer_s13Q · 2022-10-23

**Confidence:** 4
**Clarity, Quality, Novelty And Reproducibility:** Good.
**Correctness:** 3
**Technical Novelty And Significance:** 3
**Empirical Novelty And Significance:** Not applicable
**Recommendation:** 5

**Strength And Weaknesses:**

I feel this is an interesting paper. Considering testing time interaction is the right way to think about generalization in RL. The lower bound in Section 4.1 is not surprising and intuitive. I am glad the author can make it formal. I have two major concerns about this work:

1. The authors mentioned in the training stage, the agent can sample i.i.d MDP instances from the unknown distribution. I can not understand why the agent has such power. This can only happen in a highly-simulated environment and I am not aware any real-application can satisfy this requirement. Algorithm 1 needs to sample a large number of MDPs. If the state space is infinity, how can the agent do that? For example, if I want to train an agent to play Atari, how Algorithm 1 samples MDPs?

2. Theorem 2 is less satisfactory, especially for C(D) term. This complexity measure is not interpretable and so far it's only better than SA in a very extreme case. I feel this should explicitly depend on some quantities that describe the distribution of D. Usually for a tabular MDP, people put Dirichlet prior on the dynamic, which is continuous. Then C(D) should depend on the Dirichlet parameters. It's unclear how good C(D) is and how it reflects the intermediate regime.

In the end of Section 4.2.2, the author mentions "when D is subgaussian or mixtures of subgaussian". What does it mean? The dimension of a MDP is a SxAxS tensor then how do you define sub-gaussian here?



**Summary Of The Paper:**

This paper studied the generalization problem in RL and used Bayesian regret as a performance measure. The authors considered if the agent is allowed to use the data from interaction with the testing environment.

**Summary Of The Review:**

This is an interesting paper but the theory part is less strong.

---

> ### Author Response · Authors · 2022-11-10
> **Reply to Reviewer s13Q**
>
> We sincerely thanks for your careful and insightful reviews. Here we reply your concerns separately.
> - Sampling MDPs: The ability of sampling MDPs from the unknown distribution has also been required in many previous works studying RL generelization (e.g. [1]), which we believe is a neccessary condition to make RL generalization tractable. The sampling oracle can be achieved in many RL generalization applications such as video games and robotics with a simulator. For example, in video games like Atari, we can sample the random seed of the game to generate an MDP instance. In robotics with a simulated environment, we can sample the hidden parameter of the simulator to generate a set of MDPs in the pre-training stage, and fine-tune the policy in the real-world interactions.
>
>
> - The number of samples when the state space is infinite: The sample complexity of our Algorithm 1 does not depend on the size of state and action space. Therefore, even if the state space is infinite, the number of samples we need is still bounded by $O(\frac{\mathcal C(\mathbb D)}{\delta^2})$ with high probability.
> - Theorem 2: The distribution-dependent measure $\mathcal{C}(\mathbb{D})$ can be small when the prior distribution $\mathbb{D}$ can provide enough information. Obviously, when the distribution $\mathbb{D}$ is uniform over the set of all tabular MDPs, we cannot expect pre-training to improve the test regret, since it provides no extra information for the test stage. As discussed at the end of the Section 4.2.2, $\mathcal{C}(\mathbb{D}) \ll SA$ when $\mathbb{D}$ is supported on a small MDP set $\Omega$, or when $\mathbb{D}$ concentrates on a  sub-region of its support $\Omega$.
> - The mixture of Subgaussian: By saying $\mathbb{D}$ is subguassian or mixture of subguassian, we mean that $\mathbb{D}$ is a subguassian distribution or mixture of subguassian distribution defined on a discrete MDP set $\Omega$, instead of all possible $S\times A \times S$ tensors. One example in our paper is a Possion Distribution with infinite support $\mathbb N$ and the MDP $\mathcal M_k$ has p.d.f. $p(k) = \frac{\lambda^k e^{-\lambda}}{k!}$.
>
>
> [1] Malik et al. When Is Generalizable Reinforcement Learning Tractable? NeurIPS 2021.

---

> > ### Author Response · Authors · 2022-11-17
> > **Response to Reviewer s13Q**
> >
> > Dear Reviewer s13Q, we would be grateful if you could check whether our response has addressed your concerns and let us know if any issues remain. To recap our response,
> > - We clarify that our upper bound is an improvement to previous works in many cases. To address your concern about why we cannot expect a better bound in other cases, we also append a lower bound in Appendix C.4. Please check the appendix with red color text.
> > - We address your concern about the power of sampling MDPs.

---

> > ### Comment · Reviewer_s13Q · 2022-11-17
> > **Thanks for the response**
> >
> > Thanks for the response. Can you please provide a detailed description on how your algorithm could work for Atrai environment, saying Ms. Pacman? I think Atrai simulator is a deterministic environment except for the initial state. Then how you sample the random seed of the game to generate an MDP instance? This is also asked by Reviewer tCeD.

---

> > > ### Author Response · Authors · 2022-11-18
> > > **Thanks for the prompt reply**
> > >
> > > Thanks for your prompt reply. Before answering your question, we want to highlight that this is a theoretical work toward understanding the power of pre-training in RL generalization. We aim to better understand RL generalization instead of designing algorithms for specific tasks. This also follows the research style of RL generalization in the previous works (e.g. [1,2]).
> > >
> > > Atari games such as Ms. Pacman are proposed for RL algorithms *in the single-task setting*, and the algorithm is evaluated by the performance of the single task. That is, this benchmark can not be directly used to study generalization in RL. Nevertheless, to study RL generalization from an empirical perspective, some works have recently extended the Atari games to the multi-task generalization setting (e.g., Procgen), in which the algorithm is measured by the performance of the sampled MDP from the task distribution in the test stage. We can sample the random seed of the game to generate new MDP instances in these benchmarks. In this case, the intuitive implementation of our algorithm can be described as follows. In the pre-training stage, we sample several MDPs and learn a policy set that can perform well in these MDPs using single-task RL algorithms. Then in the test stage, we find the best policy in the policy set via several test-stage interactions.
> > >
> > > [1] Malik et al. When is generalizable reinforcement learning tractable? NeurIPS 2021
> > >
> > > [2] Wang et al. On the generalization gap in reparameterizable reinforcement learning. ICML 2019

---

### Official Review · Reviewer_tCeD · 2022-10-24

**Confidence:** 3
**Correctness:** 4
**Technical Novelty And Significance:** 2
**Empirical Novelty And Significance:** Not applicable
**Recommendation:** 5

**Clarity, Quality, Novelty And Reproducibility:**

The paper is clearly written, and the theorems are well-presented and contextualized. The derived regret bounds in the particular pretraining setting with test-time interaction considered by the authors are novel to the best of my knowledge.

**Strength And Weaknesses:**

Strength
- The paper offers a more precise understanding of the best-case benefits of pretraining over no pretraining when fine-tuning is allowed for $K \rightarrow \infty$ episodes at test-time.
- Extensive theoretical investigation of the benefits of pretraining in RL with the help of two proposed algorithms for the setting with finetuning or test-time interaction in the non-asymptotic case and without further test-time interactions.
- The authors do a good job at contextualizing and explaining the theoretical implications and ramifications of their derived bounds.
- The paper is well-structured and clearly written.
- Appropriate literature review to contextualize the contributions

Weakness
- While I appreciate the extensive analysis and theoretical nature of the proposed algorithm my biggest concern is that the benefits and usefulness of these bounds in practical pretraining settings in RL are not very clear from the paper. I would have hoped the authors could at least implement their algorithm and test how well it actually works in popular RL environment test beds, e.g. the Arcade Learning Environment. Without such experiments or empirical results, my worry is that the algorithms and bounds are of limited use. As an example, how effective and computationally expensive would the algorithm be in practice over naively pretraining an agent on the subset of MDP environments, and then adapting to a new environment of the same distribution?
- The proposed bounds and algorithms assume the pretraining and test environments are sampled iid while pretraining is especially hoped to help in OoD scenarios.


**Summary Of The Paper:**

This paper theoretically investigates the implications of pre-training for generalization in RL in terms of possible benefits and limitations. Specifically, the paper shows that in an asymptotic limit (where the agent can interact $K \rightarrow \infty$ times with its environment and knows the MDP distribution D) pre-training can improve the regret only up to a constant. In the non-asymptomatic case, the authors propose an algorithm for which pretraining can help, by deriving a regret bound that depends on the complexity of the (MDP) distribution D. Finally, authors propose a second algorithm that pursues optimalitiy in expectation for the case where the agent is not allowed for further test-time interactions after pretraining.

**Summary Of The Review:**

This paper offers a range of novel theoretical insights regarding the benefits (or impossibilities) of leveraging pretraining for generalization in RL where the pretraining and test environments are sampled from the same distribution. While the paper is purely theoretical and I believe such insights to be helpful in general, it is unclear how much those particular insights help in practice where pre-training is already a frequently used technique for RL. I would have expected the authors to at least empirically prove the effectiveness of the suggested algorithms on RL test beds. I, therefore, lean towards rejecting the paper.

---

> ### Author Response · Authors · 2022-11-10
> **Reply to Reviewer tCeD**
>
> We sincerely thank you for your careful and insightful review. Here we reply your concerns separately.
> - The benefits for the practical settings: This work aims to understand RL generalization from the theoretical perspective and provide practical insight for RL generalization. Specifically, our algorithms and theorems provide the following takeaways for practical settings:
> 1. In general, we systematically analyse the difference of generalization in suppervised learning and RL, and provide a theoretical separation between optimalily *in expectation* and optimaly *in instance*. We show that optimality in instance can not be obtained without further interactions with the test environment (Proposition 1). This implies that fine-tuning is neccessary once we pursue a near-optimal policy in the test environment.
> 2. The lower bound (Theorem 1) shows that that the benefit of pre-training can be limited in the asymptotic setting. As the number of test-time interaction increases, the information provided from the prior distribution $\mathbb{D}$ can gradually vanish.
> 3. One possible way to gather pre-training information is to maintain a diverse policy set that can perform well in the MDPs sampled in the pre-training stage. Our theorem (Theorem 2) shows that this policy set captures important knowledge of the MDP distribution, and it can reduce the regret in the test stage. This idea can possibly be applied in the algorithm design for the RL generalizations.
>
> - OOD scenarios: Thanks for the comments. As discussed in the Conclusion section, out-of-distribution generalization in RL under certain distribution shifting is an interesting problem remaining unclear. However, OOD generalization theory is still developing even in traditional supervised learning [1]. Since the understanding about RL generalization is still initial and incomprehensive, we believe it is better to start from the i.i.d. setting.
>
> [1] Gulrajani I, Lopez-Paz D. In Search of Lost Domain Generalization. ICLR 2020.

---

> > ### Author Response · Authors · 2022-11-17
> > **Response to Reviewer tCeD**
> >
> > Dear Reviewer tCeD, we would be grateful if you could check whether our response has addressed your concerns and let us know if any issues remain. To recap our response,
> > - We summarize three takeaways from our theoretical results that can benefit practical RL algorithm design. Specifically, the takeaways include the recognition of optimality *in instance* and *in expectation*, the limitation in the asymptotic setting, and the idea of maintaining a representative policy set that can generalize well.
> > - We address your concern about RL generalization in the ID and OOD settings.

---

> > > ### Comment · Reviewer_tCeD · 2022-11-21
> > > **Response**
> > >
> > > I want to thank the authors for their response. However, I am still not fully satisfied regarding the practicality and usefulness of these theoretical results in any practical setting given the lack of empirical experiments to support the claimed benefits. I, therefore, keep my rating for the moment and believe this work would greatly benefit from actual experiments that prove the practicality or usefulness of their algorithms.

---

### Official Review · Reviewer_SxSR · 2022-10-27

**Confidence:** 3
**Correctness:** 3
**Technical Novelty And Significance:** 3
**Empirical Novelty And Significance:** Not applicable
**Recommendation:** 8

**Clarity, Quality, Novelty And Reproducibility:**

The presentation is overall confusing, and it does not help evaluating the contribution of this paper. Notation is not always sharp or intuitive, some symbols are never defined, there are a few sloppy claims and confusing comments. Presentation issues include:
- The paper often refers to "learning/knowing $\mathbb{D}$", but the algorithms actually learn set of policies and not the task distribution;
- The cardinality of the set of MDPs is denoted sometimes with $\Omega$ and sometimes with $M$ (I am not sure the latter symbol have been introduced actually);
- Algorithm 1 reports a symbol $\mathcal{U}$ that is defined in the appendix. The Subroutine 4 is only reported in the appendix though it is crucial to understand the algorithm;
- It is not clear how many times the loop at line 3 of Algorithm 1 has to be executed.
- "We select $(\pi_l, v_l)$ with the most optimistic value $v_l \in \Pi_l$". It is not clear how the value can be (optimistically) estimated without knowing the true test MDP;
- "$\mathcal{C} (\mathbb{D})$ is still bounded". Not clear why.

**Strength And Weaknesses:**

Strengths
- (Relevance) The paper addresses the interesting problem of pre-training a set of policies for efficient RL at test time;
- (Originality) The paper introduces several original ideas, including interesting problem formulations and algorithms to pre-train policy sets.

Weaknesses
- (Presentation) The paper is not an easy read. The notation is not always sharp, the claims are sometimes sloppy and not clearly supported, and some of the comments on the theoretical results can be confusing;
- (Odd evaluation choices) The benefit of pre-training is studied either in a bizarre asymptotic formulation of the regret or compared against a RL baseline that learns from scratch neglecting the samples complexity of the pre-training itself;
- (Unreasonable results) Some of the theoretical statements are so strong that look unreasonable (though I am probably missing something);
- (Non-asymptotic lower bound) There is a clear mismatch between the provided lower bound (on the asymptotic cumulative regret) and the regret upper bounds of the proposed algorithms, which instead come from non-asymptotic analysis;
- (Related works) Only a few related works are reported, and the relations with some relevant pieces of the literature, such as multi-task RL (e.g., Brunskill & Li, Sample complexity of multi-task reinforcement learning, 2013), are not discussed well-discussed.

Comments

(Lower bound) I am wondering what is the purpose of the asymptotic setting. Regret minimization is usually interesting in finite time, what is the point of comparing the infinite values of the asymptotic cumulative regrets? Moreover, all of the other results are reported on finite-time settings, for which a lower bound is not formally provided. This makes unclear how to evaluate the obtained upper bounds.

(Measure of performance) I am not sure the test-time regret alone is a good measure to uphold the importance of pre-training. It is clear that pre-training can provide some benefits at test time, but what is the point in comparing the $O(\sqrt{SAHK})$ regret of RL from scratch with the $O(\sqrt{\mathcal{C}(\mathbb{D}) K})$, when the PCE algorithm still suffers $O(|\Omega| \sqrt{SAHK})$ regret in the pre-training phase (or, alternatively, sample complexity)?

(Algorithm 1) Is the PCE algorithm always guaranteed to find a suitable set of policies during pre-training? I.e., the condition at line 10 is guaranteed to be verified eventually?

(Theorem 3) I am in all likelihood missing something here, but the result in Theorem 3 seems to be so strong to be unreasonable. In particular, when the covering number on the policy space is finite the output of OMERM can generalize to any unseen task without further interactions. The following thought experiment suggest otherwise: If I sample a test task from $\mathbb{D}$ and then a policy uniformly at random from the obtained policy set, in all likelihood the policy will not be good for the specific instance. If I repeat this process several times (then converging to the expectation) I cannot see how the average sub-optimality can be small. What is wrong with my thought experiment?

(Non-Markovian policies) The paper seems to be only considering Markovian policies, but I am wondering whether this setting necessitates history-based policies. From my intuition, the test task is akin to a partially observable MDP, where the information on the specific instance is hidden to the agent. Do the authors think that Markovian policies are sufficient instead?

**Summary Of The Paper:**

This paper addresses a generalization problem in RL, in which the agent interacts with a set of MDPs taken from an unknown task distribution during training, and subsequently minimizes the (expectation) of the regret in a test MDP taken from the same distribution. The paper first studies an asymptotic setting in which the number of test episodes goes to infinity, showing that the benefit of pre-training is at most a constant factor. Then, it considers a non-asymptotic setting through an algorithm, called PCE, that pre-trains a set of policies to achieve a regret upper bound that depends on the complexity of the task distribution but not on the number of states and actions in the MDPs. Finally, the paper studies an additional setting in which no interactions can be taken at test time, for which it provides an algorithm (OMERM) achieving near-optimal zero-shot adaptation to the test MDP by taking a polynomial number of samples during training.

**Summary Of The Review:**

*After discussion*

After a fruitful discussion with the authors, I have a better understanding of the contributions of the paper. Most of my concerns have been solved, and I am updating my evaluation to accept.

---


This paper is in many ways interesting. Generalization in RL is an extremely relevant problem, while the algorithmic procedures and the problem formulations seem to introduce original ideas. Especially, the idea of achieving guaranteed test-time sub-optimality/regret by covering the space of instance-optimal policies looks very promising to me.

However, the presentation is less than ideal, and as a result there is a good chance I am missing crucial parts of this submission. The reported results seem sometimes unreasonable to me (especially Theorem 3), and I cannot see how this problem can be solved without strong structural assumptions on the task distribution and the underlying MDPs.

I am currently providing a borderline evaluation that reflects my skepticism about some of the claims. I will revaluate my score upon a due inspection of the proofs, which I could not check given the limited reviewing window, and clarifications from the authors' response.

---

> ### Author Response · Authors · 2022-11-10
> **Reply to Reviewer SxSR (Part Ⅰ)**
>
> We sincerely thanks for your careful and insightful review. We hope the following replies can help with clarification and are eager for more discussion.
>
> - Lower Bound:
> 	- The purpose of the asymptotic lower bound is to capture to what extent pre-training could possibly be useful, or in other words, how much information the pre-training stage can provide at most. The lower bound shows that the information provided from the distribution $\mathbb{D}$ can be gradually useless as $K$ increases asymptotically.
> 	- In response to your comment on the non-asymptotic regime, we also add a non-asymptotic test-time regret lower bound $\Omega(\sqrt{\mathcal C(\mathbb D)K})$ in Appendix C.4 to indicate that the test-time regret in Theorem 2 is tight. This bound matches our upper bound except for logarithmic factors. Thanks for the helpful suggestion, and please find it in the modified paper.
>
> - Measure of performance: The sample complexity in the pre-training stage can be captured by the number of queries of the oracle $\mathbb{O}_l$, which depends linearly on $\mathcal{C}(\mathbb{D})$. The reason why we mainly focus on the test-time regret is that the samples in the pre-training stage are relatively more sufficient and easy to obtain in many RL generalization applications. One motivating example is the robotics task where we have a simulator to efficiently and safely generate unlimited data. The reason to compare the test-time regret with the regret from scratch is to show the benefits of the pre-training. With the information provided in the pre-training stage, we show that the dependence on $S, A$ can be reduced to the distribution complexity measure $\mathcal{C}(\mathbb{D})$.
>
>
> - Algorithm 1: By Lemma 10 in Appendix C.2, we show that the condition at line 10 will be satisfied within $\log (\tilde{\mathcal{O}}(\mathcal{C}(\mathbb{D})))$ phases with high probability ($1-\delta$). Since $\delta$ is of order $\tilde{\mathcal{O}}(1/ \sqrt K)$ and the regret in the test MDP is at most $K$, the influence to the total regret is still negligible even when this high probability event does not hold.
>
>
> - Theorem 3: We want to clarify that in Theorem 3 we consider the near-optimal policy *in expectation*, i.e. near-optimal w.r.t. the policy $\pi^*(\mathbb D) = \arg\max_{\pi \in \Pi} \mathbb E_{\mathcal M \sim \mathbb D}V_{\mathcal M}^\pi$. That is, we do not require the returned policy to be near-optimal w.r.t. the optimal policy of the specified sampled MDP instance (optimality in instance). As we show in Proposition 1, it is impractical to demand optimality *in instance* when interactions with the test MDP are now allowed. The basic idea of Algorithm 2 is to find a policy that empirically performs well in the sampled $N$ MDPs in an average manner. Once $N$ is sufficiently large, the empirically good policy can be guaranteed to be near-optimal w.r.t. $\pi^*(\mathbb D)$.
>
>
> - Non-Markovian policies: Thanks for the valuable comments. The perspective of Non-markovian policies and POMDPs is also an interesting direction to understand RL generalization. However, solving a POMDP typically requires exponential sample complexity without additional assumptions [1,2], which has not been well-understood in the RL community. In this paper, we study RL generalization from the MDP formulation. We leave the POMDP perspective as an interesting future direction.
>
> - Related work on multi-task RL: Thanks for the comment. We have discussed the related work on multi-task RL at the end of the first paragraph in the Related Work section, but neglect to cite the work of Brunskill & Li, 2013. The complexity bounds in Brunskill & Li, 2013 are derived under additional assumptions such as a known gap of model difference in the MDP set, and are different from ours. Moreover, their problem formulation is different from ours since they study the multi-task setting where the MDP is selected from a given MDP set. we added this to the Related Work section in our modified paper.
>
> [1] Jin et al. Sample-efficient reinforcement learning of undercomplete pomdps. NeurIPS 2020
>
> [2] Kwon et al. RL for Latent MDPs: Regret Guarantees and a Lower Bound. NeurIPS 2021

---

> > ### Author Response · Authors · 2022-11-10
> > **Reply to Reviewer SxSR (Part Ⅱ)**
> >
> > We also thank you for your comments on the clarity. We will revise the paper accordingly, and below are our replies:
> > - Learning/knowing $\mathbb D$: We mainly refer to "learning/knowing $\mathbb D$" in Section 4.1, in which we provide a lower bound showing the limited benefit of learning/knowing the distribution $\mathbb D$ exactly. We do not require our algorithm in Section 4.2 to learn the exact distribution $\mathbb D$ in the pre-training stage.
> >
> > - The notation $\Omega$ and $M$: We use $\Omega$ to denote the MDP set, and use $M$ to denote the cardinality of $\Omega$. We will clarify the notation in the paper.
> > - The symbol $\mathcal{U}$ and the Subroutine 4: The symbol $\mathcal{U}$ is only used in the appendix. The Subroutine 4 is a technical algorithm that selects a policy set that covers $(1-3\delta)$-fraction of the MDPs in the sampled MDP set. The basic idea of subroutine 4 is to gradually select policies with maximum number of the covered MDPs. We organize it in the appendix mainly for cleaness.
> > - The number of execution: As stated by Lemma 10 in Appendix C.2, the loop at line 3 of Algorithm 1 is executed at most $\log (\tilde{\mathcal{O}}(\mathcal{C}(\mathbb{D})))$ times with high probability.
> > - The definition of $v_l$: Note that $\hat{\Pi}$ is a set containing policy-value pair $(\pi, v)$, where $\pi$ is a near-optimal policy for a certain MDP $\mathcal{M}$, and $v$ is the evaluated value of $\pi$ on the MDP $\mathcal{M}$. Both $\pi$ and $v$ are calculated in the pre-training stage using the oracle $\mathbb{O}_l$ and $\mathbb{O}_e$. Therefore, we do not need to evaluate $v$ in the fine-tuning stage.
> > - $\mathcal C(\mathbb D)$ bounded: When $|\Omega|$ is large, the distribution complexity $\mathcal{C}(\mathbb{D})$ is still bounded when $\mathbb{D}$ enjoys certain benigh properties. Specifically,  consider a Possion Distribution with infinite support $\mathbb N$ and p.d.f. $p(k) = \frac{\lambda^k e^{-\lambda}}{k!}$. In such case, $\mathcal C(\mathbb D) = O(\lambda \ln \frac{1}{\delta})$ is still finite.

---

> > > ### Author Response · Authors · 2022-11-17
> > > **Response to Reviewer SxSR**
> > >
> > > Dear Reviewer SxSR, we would be grateful if you could check whether our response has addressed your concerns and let us know if any issues remain. To recap our response,
> > > - We clarify the purpose of our asymptotic lower bound, and to address your concern, we append a non-asymptotic lower bound in the appendix.
> > > - We clarify the rationality of Theorem 3 by pointing out that we are pursuing a near optimal policy *in expectation* rather than *in instance*.
> > > - We address your concern on algorithm 1 (PCE), and explain why we focus on the test time regret.

---

> > > > ### Comment · Reviewer_SxSR · 2022-11-17
> > > > **After Response**
> > > >
> > > > I want to thank the authors for addressing my concerns with patience and clarity, and for updating the paper following reviewers' suggestions.
> > > >
> > > > Thanks to their replies, I believe to have now a better picture of the contributions of this paper and the significance of the reported results. Especially, the addition of the non-asymptotic lower bound is further upholding the value of the PCE algorithm and its analysis, and the result of Theorem 3 is completely reasonable as it refers to near-optimal policy in expectation over D.
> > > > Moreover, other reviewers did find the paper sufficiently clear, which makes the presentation of the paper less concerning.
> > > >
> > > > For these reasons, I am willing to update my score upwards. In my opinion, the paper could be even more valuable with:
> > > > 1) An additional result showing whether Markovian policies are sufficient for this setting (even if the authors are rightly noting that learning non-Markovian policies is likely intractable anyway);
> > > > 2) An extended analysis that considers non-zero costs for pre-training.
> > > >
> > > > As a final note, while the result in Theorem 3 is now clear, I am wondering whether the near-optimality in expectation is really a sensible target, as opposed to learning a policy that allows for efficient (few-shot) adaptation to the near optimal policy in instance.

---

> > > > > ### Author Response · Authors · 2022-11-18
> > > > > **Thanks for the positive comments**
> > > > >
> > > > > Thanks for the valuable suggestions that could make our paper even more valuable! We will update the paper accordingly. Specifically,
> > > > > - Whether Markovian policies are sufficient: Our Theorem 2 states that the regret in the test time is $\tilde{O}(\sqrt{\mathcal{C}(\mathbb{D})K})$. From a standard regret-to-PAC conversion, our test-stage algorithm can be transformed to an algorithm that returns an $\epsilon$-optimal Markovian policy with $\tilde{O}(\mathcal{C}(\mathbb{D})/\epsilon^2)$ test-stage trajectories. That is, a Markovian policy set suffices to perform near-optimal in the sampled test environment.
> > > > > - Non-zero costs for pre-training: In this setting, one possible measure of the cost in the pre-training stage is the number of queries of our learning oracle, which has a linear dependence on the distribution complexity $\mathcal{C}(\mathbb{D})$ in our Algorithm 1. Thanks for pointing out this, and we will add a discussion in the final version.
> > > > > - Near-optimality in expectation: We agree with the reviewer that near-optimality *in instance* is a more valuable and interesting setting for RL generalization, which is also our main focus in this work. We study near-optimality *in expectation* because it is in line with generalization in supervised learning, and it separates the generalization problem in supervised learning and RL.
> > > > >
> > > > > Thanks again for you positive comments and raising the score!

---

### Decision · Program_Chairs · 2023-01-20

**Decision:**

Reject

**Justification For Why Not Higher Score:**

Missed connection and comparison to important related work. Missed empirical validation of the applicability of the proposed algorithm.

**Justification For Why Not Lower Score:**

N/A

**Metareview: Summary, Strengths And Weaknesses:**

The paper studies the theoretical benefits and limitations of pre-training on generalization of RL. It shows that asymptotically, the improvement from pre-training and knowing the MDP distribution D is at most a constant factor when the agent can interact with the environment at test time. The authors then designed algorithms in the non-asymptomatic case, for which pre-training helps and  derived a regret bounds depending on the complexity measurement on the MDP distribution D.

 The reviewers remained split after the rebuttal and offline discussion. One reviewer consider it an important theoretical work "establishing (previously unknown) statistical barriers on RL generalization across environments", while the other reviewers remain skeptical on the "applicability and usefulness of the presented algorithms". The paper follows closely prior work on "When Is Generalizable Reinforcement Learning Tractable?", which also 1) studies generalization of RL across environments, 2) points to the conclusion that "tractable generalization is impossible in the worst case" despite strict conditions on the environments, and 3) proposes algorithms that improves generalization under strong proximity condition. Unfortunately the authors only mentioned the prior work in related work section and fail to address the connection and difference of the proposed work vs prior work. IIUC, the paper designed algorithms and derived regret bound under weaker conditions on the environments. But it is hard to compare generalization of the proposed algorithms vs prior work say under the same condition. Without properly setting the context and comparing to the related work, it is hard to gauge how much additional theoretical value is provided. With the additional caveat of missing any empirical validation of the proposed algorithms, we are not comfortable accepting the paper as is.